# Coseismic fault sealing and fluid pressurization during earthquakes

Lu Yao [1] ✉, Shengli Ma[1] & Giulio Di Toro [2,3]

Earthquakes occur because faults weaken with increasing slip and slip rate. Thermal pressurization (TP) of trapped pore fluids is deemed to be a widespread coseismic fault weakening mechanism. Yet, due to technical challenges, experimental evidence of TP is limited. Here, by exploiting a novel experimental configuration, we simulate seismic slip pulses (slip rate 2.0 m/s) on dolerite-built faults under pore fluid pressures up to 25 MPa. We measure transient sharp weakening, down to almost zero friction and concurrent with a spike in pore fluid pressure, which interrupts the exponential-decay slip weakening. The interpretation of mechanical and microstructural data plus numerical modeling suggests that wear and local melting processes in experimental faults generate ultra-fine materials to seal the pressurized pore water, causing transient TP spikes. Our work suggests that, with wear-induced sealing, TP may also occur in relatively permeable faults and could be quite common in nature.

Dynamic weakening of seismic faults is essential for the generation of large earthquakes[1,2]. The underlying mechanisms responsible for the dynamic fault weakening are diverse, but largely are thermal in origin[3–5]. Some of the proposed weakening mechanisms, such as frictional melting and flash heating, have been corroborated successfully by high-velocity friction experiments[6,7]. Yet thermal pressurization (TP), which is considered as a widespread process for fault weakening during earthquakes from theoretical analysis[8,9], numerical modeling[10–13], and seismological studies[14], still lacks solid experimental evidence. Some recent experimental work questions about the efficiency of TP in cohesive fault rocks[15], and a few more demonstrate that TP could be significantly counteracted by shear-induced dilatancy at least during the early stage of seismic slip[16–18]. Yet experimental observations of close relation between fault weakening and rock permeability support TP being a dominant weakening mechanism[19]. With these contradictory results from laboratory experiments, the activation of TP and its role in earthquake faulting remains enigmatic.

Large earthquakes nucleate at a small fault patch and occur as dynamic frictional ruptures propagating along preexisting faults, with the pulse-like rupture mode[20] in many cases (Fig. 1a). During the short period (much shorter than the event duration) slip at a given fault patch, the physico-chemical processes within the slipping zone can be

quite complicated. For a mode II shear rupture propagating in a water-saturated fault patch within silicate rocks at depth, the scenario could be (in chronological order; Fig. 1b): (1) fracturing and fragmentation occur due to the increased stress level near the rupture tip (asymmetric cracks may form); (2) flash melting occurs on asperity contacts as frictional slip commences; (3) fragmentation and crushing continue, intense wear occurs, discontinuous melt patches form, and thermal pressurization of pore water is activated; (4) bulk melting occurs as pore water escapes from the slipping zone and a continuous molten layer may form if slip displacement is large enough. This scenario has been partly suggested by previous studies[21–23]. However, how the involved processes especially those listed in (3) and (4) may interact in the presence of water to affect fault weakening has been virtually unexplored.

Here, we examine the TP process and its possible interaction with wear and local and bulk melting through high-velocity (equivalent slip rate[24] $V_{eq} = 2.0$ m/s) friction experiments performed on dolerite under pore water pressure ($P_p$) up to 25 MPa. We observe transient TP weakening in dolerite-built faults that initially have open boundaries for fluid flow. These results together with microstructural and modeling data suggest that sealing induced by wear and local melting may compete with fracturing and dilatancy to affect the fluid

[1]State Key Laboratory of Earthquake Dynamics, Institute of Geology, China Earthquake Administration, Beijing, China. [2]Dipartimento di Geoscienze, Università di Padova, Padova, Italy. [3]Istituto Nazionale di Geofisica e Vulcanologia, Rome, Italy. ✉e-mail: luyao@ies.ac.cn

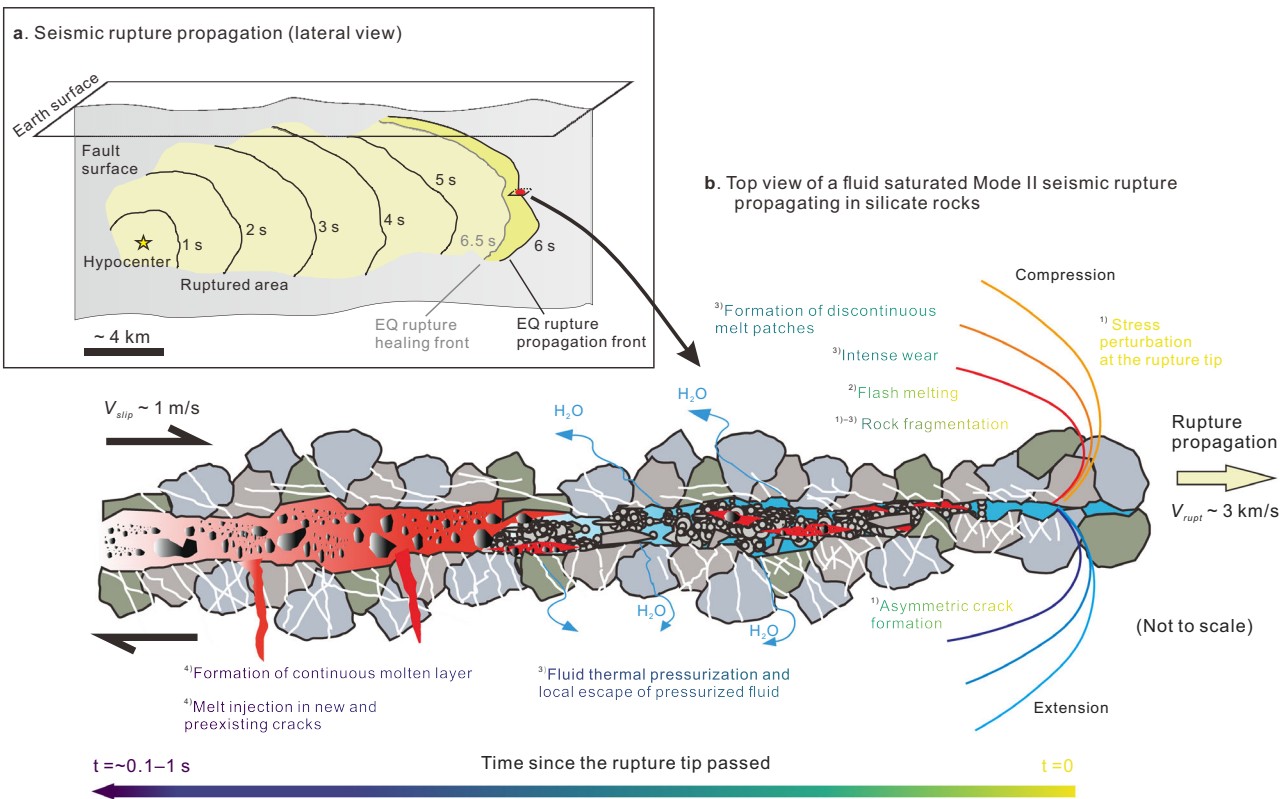

**Fig. 1 | Conceptual model of seismic rupture propagation. a** Earthquake rupture propagation along a fault as a self-healing pulse. **b** Seismic slipping zone at the scale down to microns with the sequence of deformation events triggered by the passage of the rupture propagation front and frictional heating at its wake in a fluid-saturated fault (modified from ref. [21]). Figure 1b is drawn not to scale: the thickness of the molten layer, the size of the surrounding mineral grains, and the length of the slipping fault patch, are a few mm to cm, a few μm to mm, and a few hundred to thousand m, respectively.

transport properties and promote TP weakening of faults during seismic slip.

## Results

### High-velocity friction experiments

We perform experiments using a rotary-shear low- to high-velocity friction apparatus[25] equipped with a newly-conceived pressure vessel (see "Methods" and Supplementary Fig. S1a, b). Deionized water was used as pore fluid in most of the experiments (except for one run in which high-purity nitrogen filled the entire vessel), although the pore pressure was exerted by the pressurized nitrogen at the upper part of the vessel in all the experiments. The cylindrical dolerite specimens were prepared with flat or pitted end surfaces to vary the amount of pore water in the fault slip zone (referred to as flat and pitted slip surfaces, respectively; see Supplementary Fig. S1c, d). In the experiment, pore pressure was monitored by using a pressure sensor far from the experimental fault (reflecting the bulk pore pressure), while temperature was measured by a sheathed thermocouple exposing on the slip surface (15 mm away from the center; Supplementary Fig. S1b, c).

As a reference, experiment LHV964, performed at effective normal stress $\sigma_{n\,eff} = 3$ MPa and room humidity conditions, is displayed in Fig. 2a. The friction coefficient ($\mu$) rapidly increases to a first peak of 0.52 and drops quickly to 0.35 (red line). Then $\mu$ gradually increases to a higher second peak of 0.71, followed by a nearly exponential decay to a "steady-state" friction coefficient ($\mu_{ss}$) of 0.30. The temperature (blue line) increases sharply in the first 2.0 m of slip to 855 °C and then gradually up to 1250 °C.

By contrast, the friction coefficient evolves in a more complicated manner in the experiments performed under pore fluid pressure. The post-peak friction evolution includes an initial decay to $\mu$ ~0.6,

followed by a sharp weakening stage ($\mu$ reduces to <0.2 within 0.5–0.75 m of slip; yellow shaded area in Fig. 2b and Supplementary Fig. S2a, b), a nearly frictionless stage (minimum $\mu$ is 0–0.01 at displacements between 4 to 8 m; Fig. 2b and Supplementary Fig. S2a, b) and an overall re-strengthening stage (with large fluctuations; see Supplementary Fig. S2c) with final friction comparable to the room-humidity case (Supplementary Fig. S2c). The peak temperatures prior to the sharp weakening are 823, 591, and 851 °C for the cases of 10 MPa nitrogen, 10 MPa, and 25 MPa water as pore fluids, respectively (Fig. 2b and Supplementary Fig. S2a, b, d).

A few more experiments were performed under higher $\sigma_{n\,eff} = 6$–10 MPa and $P_p = 25$ MPa. Compared with the results shown in Fig. 2b and Supplementary Fig. S2a–c, the nearly frictionless stage tends to last for a shorter slip distance with increasing $\sigma_{n\,eff}$ (Fig. 2c, d). The post-peak friction evolution is then featured by sharp weakening and rapid slip strengthening. However, these features are influenced by the geometry of the slip surface, i.e., flat or pitted surfaces (see Supplementary Fig. S1d). In several attempts, only the experiments with the pitted slip surfaces show the sharp weakening and rapid slip strengthening (solid red lines in Fig. 2c, d) at $\sigma_{n\,eff} = 6$ and 10 MPa, in contrast to the exponential-decay type weakening after the peak friction in the tests with the flat slip surfaces (dashed red lines).

It is worth noting that whenever there is a transient sharp weakening (TSW; yellow shaded area in Fig. 2b–d and Supplementary Fig. S2a, b), there is a nearly concurrent increase in the rate of pore pressure rise (green solid lines; could be much lower than that achieved in the slipping zone). Those experiments without TSW do not record abrupt changes in the bulk pore pressure (dashed lines in Fig. 2c, d).

Furthermore, there are possible relations between the TSW and the shortening rate. Compared to the experiments conducted under

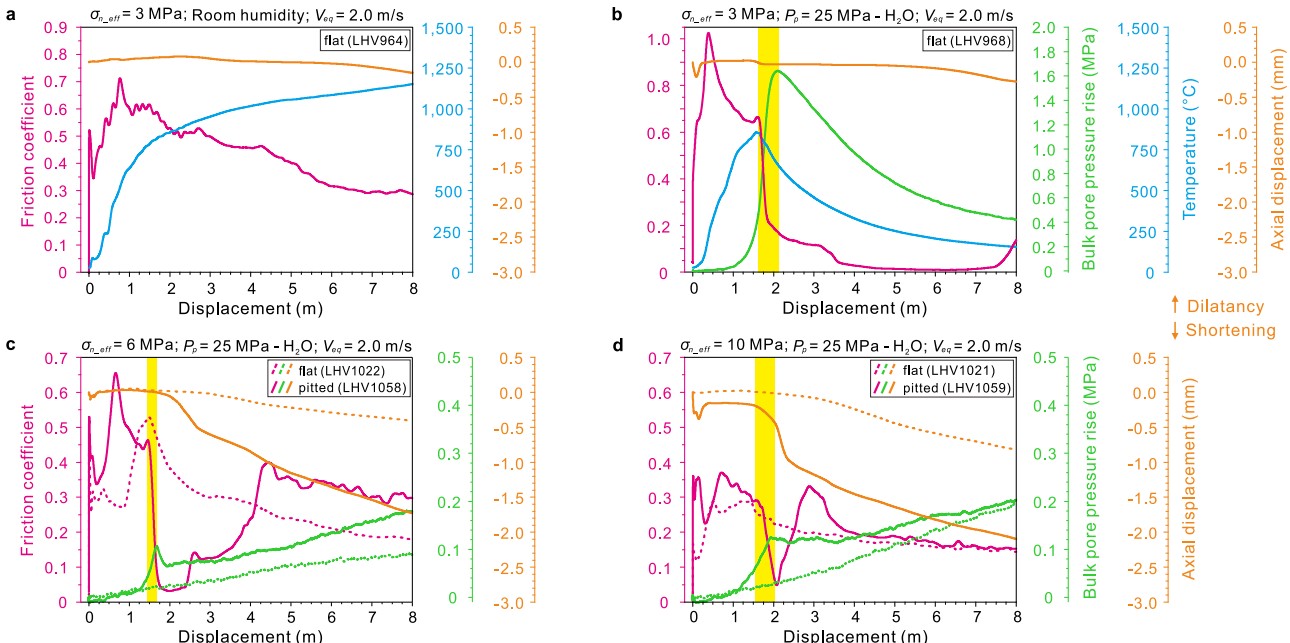

**Fig. 2 | Experimental results.** Evolution of friction coefficient, bulk pore pressure rise, temperature and axial displacement with slip displacement from experiments conducted under effective normal stress $\sigma_{n\_eff}$ = 3–10 MPa, slip rate $V_{eq}$ = 2.0 m/s, and room humidity (**a**) or pore water pressure $P_p$ of 25 MPa (**b**–**d**). Two kinds of slip surfaces, either flat or pitted (Supplementary Fig. S1d), were used in the experiments. The transient sharp weakening and the spike in bulk pore pressure coexist (yellow shading), although the bulk pressure rise should be much lower than that locally within the fault.

room humidity or those do not show the TSW under pore fluid pressure, the experiments showing the TSW always have abrupt changes in axial shortening immediately after the weakening occurs or at the onset of the re-strengthening stage, with much higher shortening rate afterwards (cp. solid and dashed lines in Fig. 2c, d and Supplementary Fig. S3a, b).

## Microstructures
We compare the microstructures of the samples retrieved from three experiments, performed under $P_p$ = 25 MPa and $V_{eq}$ = 2.0 m/s, but stopped at relatively small ($d$ = 4.5 m, $\sigma_{n\_eff}$ = 3 MPa, LHV1103; at the early stage of the TSW; Supplementary Fig. S4) and large displacements ($d$ = 10 m, $\sigma_{n\_eff}$ = 6 MPa, LHV1058; $d$ = 20 m, $\sigma_{n\_eff}$ = 3 MPa, LHV968; both tests experienced the TSW followed by slip strengthening and slight weakening; Fig. 2c and Supplementary Fig. S2c), respectively. The sample halves are not welded together after 4.5 m of slip. The slip surface is rough due to serious comminution and wear, and continuous ring-shaped slickensides are developed in some regions (Fig. 3a and Supplementary Fig. S5a). At higher magnifications, the slickensided surfaces can be either porous (Fig. 3b) or dense (Fig. 3d). The porous surfaces abound in stretched glass-like rods with striped structures (Fig. 3b, c), while the dense surfaces are characterized by quite homogenous glass-like films overlaid by fine particles (Fig. 3d, e). On the contrary, visible molten patches or layers of ~100 μm in thickness are developed after 10–20 m of slip (Supplementary Fig. S5b, c; the two dolerite cylinders are welded together in run LHV968, but not in run LHV1058), within which vesicles can be observed (Fig. 3f, g).

## Numerical modeling
The most intriguing observation from this study is the existence of the transient sharp weakening (TSW) stage that is concurrent with a spike in bulk $P_p$ and closely followed by an abrupt increase in shortening rate. Moreover, the TSW is more efficient with more water trapped within the fault (i.e., pitted slip surfaces), and the microstructural observations suggest that crushing and

comminution are dominant processes during the TSW. The most likely interpretation for all these observations is that the TSW is caused by transient TP.

We here built a finite element model with Comsol Multiphysics to estimate the local pore pressure rise by considering the sealing effects of fine wear products and melt patches generated due to comminution and local melting (Supplementary Fig. S6). In the model, a 50 μm thick shear zone composed of wear materials is sandwiched between a pair of lowly-permeable host rocks. Since those possible sealing materials exist in the ring-shaped slickenslided regions of the slip surfaces (Fig. 3a–e and Supplementary Fig. S5a), we assume that the ring-shaped sealing zones exist within the relatively permeable bulk shear zone in the model. Little is known about the numbers and the exact locations of the sealing zones, and especially their evolution in the experiments, we roughly evaluate their effects on the TP processes by considering one or three ring-shaped sealing zones (modeling results for these two geometrical models are shown in Fig. 4 and Supplementary Fig. S7, respectively). As the pore pressure rise before the TSW is our main concern, we use the monitored temperature in the first ~1.7 m slip to constrain the local pore pressure rise, on the assumption that some lowly-permeable fine wear particles or local melts (Fig. 3a–c) may act as sealing materials even if the bulk shear zone is relatively permeable (see "Methods").

With various combinations of permeability and porosity attributed to the bulk shear zone ($k_{bsz}$ and $\Phi_{bsz}$) and sealing zone ($k_{seal}$ and $\Phi_{seal}$) (see details in "Methods"), the modeling results give helpful hints on the interpretation of the experimental data and generally underpin the above working hypothesis. Figure 4a–d display the evolution of apparent friction coefficient ($\mu_{ap}$; see Eq. 6 in "Methods") estimated from the modeling, in which the TP problem was solved from 0 to 0.85 s (for the ~1.7 m slip; the time of peak friction is set as 0 s). In many cases, the rapid pore pressure rise may reduce the net normal stress to zero within 0.2–0.4 s, after which the activated physico-chemical process could be more complicated than the TP process considered in the modeling (e.g., the opening of the experimental fault may occur transiently). So only the $\mu_{ap}$ data with positive net normal stresses are

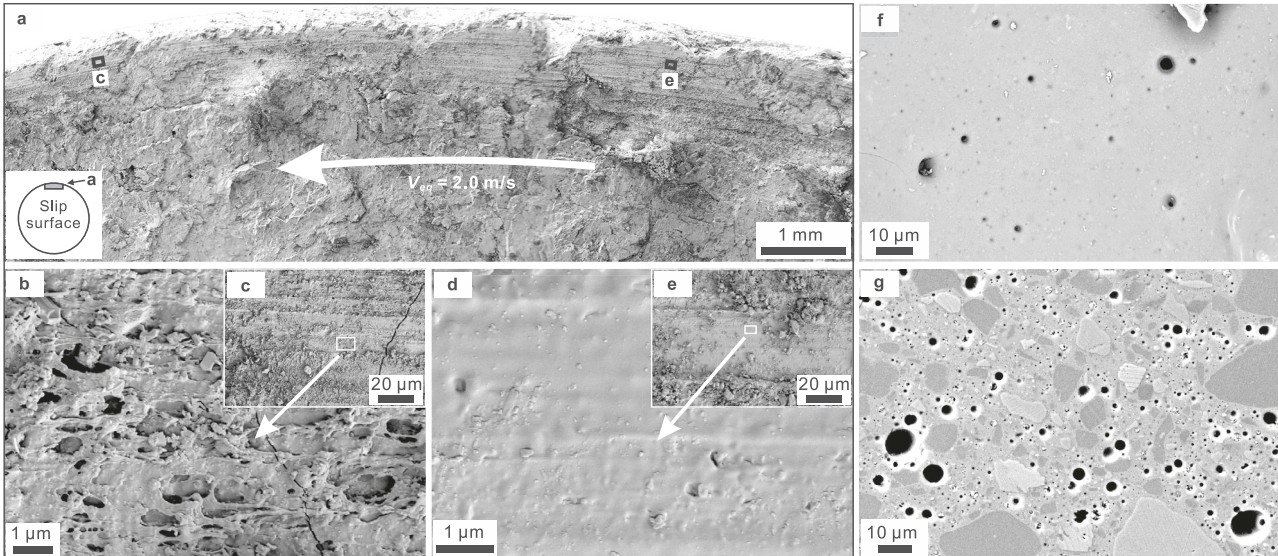

**Fig. 3 | Microstructures of the deformed dolerite samples. a–e** Secondary electron images of the slip surface retrieved from an experiment (LHV1103) stopped at 4.5 m of slip at the early stage of the transient weakening. Intense wear and local melting are dominant deformation processes, and result in porous (**b**: glass-like rods) or dense (**d**: glass-like films) slip surfaces which may affect permeability and fluid pressure rise in the slip zone. **f, g** Close-up images of the molten patches or layers developed in large-displacement (10 or 20 m) experiments (**f**: top view, LHV1058; **g**: thin-section, LHV968; see their overall textures in Supplementary Fig. S5b, c).

**Fig. 4 | Numerical modeling of sealing-enhanced thermal pressurization (TP).** The evolution of friction coefficient is predicted from the TP modeling that considers the wear-induced sealing effect (sealing zone exists in the bulk shear zone; see "Methods"). **a–d** Modeling results for various combinations of permeability ($k$) and porosity ($\Phi$) of bulk shear zone ($k_{bsz}$ and $\Phi_{bsz}$) and sealing zone ($k_{seal}$ and $\Phi_{seal}$) (see details of the model setting in Supplementary Fig. S6; some curves overlap in the figure).

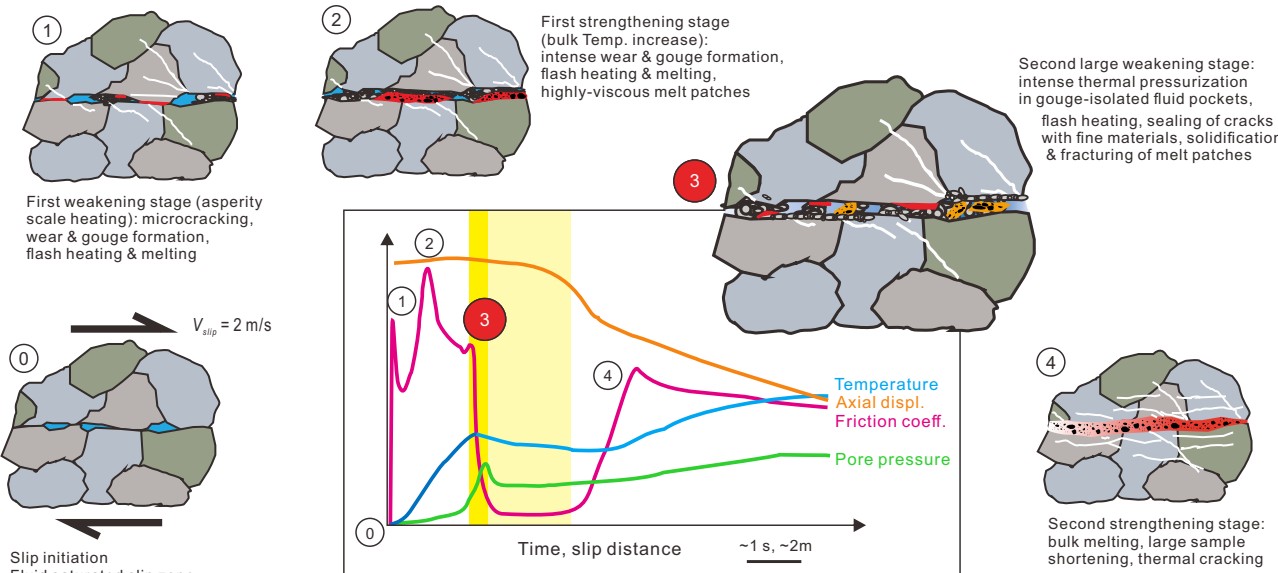

**Fig. 5 | Deformation mechanisms operating during the experiments.** This conceptual model (inspired by refs. [21–23]) links the experimental measurements of the friction coefficient, axial displacement, and temperature (Fig. 2b–d) with the microstructural observations of the experimental products (Fig. 3a–e) and the results of the numerical modeling (Fig. 4a–d). In particular, the extreme fault weakening measured in stage 3 (transient sharp weakening (TSW) plus nearly frictionless sliding; dark- and light-yellow shadings, respectively) is attributed to coseismic fault sealing or the formation of transient gouge-isolated and thermally pressurized fluid pockets.

displayed here (see the limitation of the modeling in "Methods"). Compared with the slight weakening extrapolated from the measured post-peak friction in the room-humidity experiment (also taken as the intrinsic $\mu$ in the modeling; bold gray lines in Fig. 4a–d), the TP process may cause dramatic weakening. When the entire shear zone has uniform permeability ($k_{bsz} = k_{seal}$, i.e., no sealing zone), the TP weakening becomes significant only if $k_{bsz} \leq 10^{-14}$ m$^2$ and could be negligible at higher $k_{bsz}$ (Fig. 4a; the curves for $k_{bsz} \geq 10^{-12}$ m$^2$ overlap). However, when the sealing zones have low permeability ($k_{seal} = 10^{-18}$ m$^2$), large weakening may occur even if the permeability of bulk shear zone ($k_{bsz}$) is as high as $10^{-8}$ m$^2$ (Fig. 4b; the $\mu_{ap}$ curves overlap for the cases of $k_{bsz} = 10^{-8}$–$10^{-13}$ m$^2$). With given $k_{bsz} = 10^{-15}$ m$^2$ and $k_{seal} = 10^{-18}$ m$^2$, the higher porosity of the bulk shear zone ($\Phi_{bsz}$) could yield more weakening (Fig. 4c). These could explain why the transient TP may occur on the initial bare fault surfaces and why using the pitted fault surface may increase the chances to observe the transient TP. For $k_{bsz} = 10^{-12}$ m$^2$ (permeable shear zone), $\Phi_{bsz} = 0.3$ and $\Phi_{seal} = 0.1$, significant weakening can be observed if $k_{seal} \leq 10^{-15}$ m$^2$ (Fig. 4d). Given the reported permeability data of the sheared rock samples[26], it is likely that the fine materials generated by local melting and intense wear could fulfill such a requirement of permeability (at least transiently) during our experiments.

## Discussion

The microstructural and modeling results shed light on the whole picture of friction evolution in our experiments. As previously suggested[24], the first weakening and the following strengthening are linked with flash heating and the formation of discontinuous and viscous melt patches (stages 1 and 2 in Fig. 5), respectively. In these stages, besides local melting, fine gouge particles are generated due to intense wear (Fig. 3b–e). Then the second large weakening and the following frictionless stage (stage 3 in Fig. 5) are caused by intense thermal pressurization in gouge-isolated fluid pockets in the wake of the highly-elevated bulk temperature and the activation of transient sealing process. The next slip strengthening is due to the failure of sealing and melt patches (the escape of pressurized water may increase $\sigma_{n\_eff}$ significantly) as well as the initiation of bulk melting (stage 4 in Fig. 5). Wear could be enhanced along with the transient

increase in $\sigma_{n\_eff}$, and the melt extrusion from the slip zone may come with bulk melting. The enhanced rates of losing wear and melting products from the slip zone may explain the observed changes in the shortening rate. The final steady-state friction is controlled by the shear resistance of the continuous molten layer.

Our experimental results highlight the importance of wear-induced sealing effects on the transient TP, which has been largely underestimated previously. Much attention has been paid to dilatancy and compaction in affecting the thermal[9,16,17,26,27] (and mechanical[28]) fluid pressurization. By conducting dedicated triaxial experiments with on-fault fluid pressure measurements, recent studies suggest that the dilatancy-induced pore pressure drop may counteract thermal pressurization during seismic slip[16,17]. Nevertheless, the small displacement during stick-slip in triaxial experiments may not only limit the temperature rise but also suppress the wear-induced sealing effect. Violay et al.[15] performed similar experiments as ours but they did not observe the large transient TP weakening discussed here (Fig. 2b–d). This difference may be related to the loading conditions. The high $\sigma_{n\_eff}$ (up to 40 MPa) imposed by Violay et al.[15] may facilitate the efficient transition from rock comminution and flash heating to bulk melting, so that the contribution of TP could be limited or suppressed. In addition, given the low pore fluid pressure (~5 MPa) imposed in their experiments, the tiny amount of water within the simulated fault may vaporize easily, making TP less efficient.

In natural faults, the large stress perturbation at the seismic rupture tip may result in the formation of a permeable fracture network (Fig. 1; see also in previous studies[21,29]) that cannot form in our experimental configuration (sheared pre-cut rock cylinders). However, the wear-induced sealing process active in the slipping zone may reduce the fluid flow in fault-parallel direction (transient sealed fluid pockets) and in fault-perpendicular direction (continuous ultra-comminuted and glass-like patches, Figs. 3 and 5), promoting the transient TP. Moreover, no matter how high or low the $\sigma_{n\_eff}$ might be in natural fault zones ($\sigma_{n\_eff}$ in subduction fault zones might be as low[30,31] as those in some of our experiments), the much higher degree of fault roughness in nature may result in more intense wear due to stress concentration and higher amount of water trapped within the faults, both of which may facilitate the transient TP.

The wear-induced sealing processes discussed here may also improve our understanding on the fluid-transport properties of seismic faults and their relations to TP weakening. Exploiting the mechanical conditions that lead to the formation of natural pseudotachylytes as constraints, Brantut and Mitchell[32] estimated that coseismic wall rock damage may transiently increase the fault permeability and compressibility up to ten times. However, the coseismic equivalent permeability of the fault slipping zone decreases transiently through the generation of ultrafine particles and discontinuous melt patches (Fig. 5). Clearly, fault-zone permeability during coseismic slip is a fast evolving parameter significantly affected by fracturing, gouge formation, local melting, dilatancy and gouge compaction and sealing, which cannot be evaluated properly through typical laboratory permeability measurements.

## Methods

### High-velocity friction experiments under pore pressure up to 25 MPa

The experiments were performed in a rotary-shear low- to high-velocity frictional testing machine equipped with a pressure vessel[25,33] (Supplementary Fig. S1a, b). Main parts of the vessel are shown in Supplementary Fig. S1b. The rock used is dolerite named "Fengzhen black" from Inner Mongolia, north China. The dolerite cylinders for experiments were made by using a few tool machines including coring machine, cylindrical grinder, diamond saw and surface grinder. The samples ready for the experiments are $39.98 \pm 0.01$ mm in diameter and ~20 mm in length, with the edge near one of the end surfaces cut to form two parallel surfaces for the purpose of torque transmission (Supplementary Fig. S1c). The slip surfaces were roughened with 80# silicon carbide after being leveled carefully with a surface grinder. Some of such roughened level surfaces were further pitted with small holes of 2 mm in diameter and ~1.5 mm in depth. Both of these two kinds of slip surfaces (referred to as flat and pitted slip surfaces, respectively; Supplementary Fig. S1d) were used in the experiments to evaluate the effects of the amount of water within the simulated faults on TP weakening. The existence of the pits only results in a difference in the area of slip surface by 2.25 percent, so it only exerts negligible effects on the $\sigma_{n\_eff}$ for a given normal load. To suppress the sample failure due to thermal fracturing, for each rock cylinder, an aluminum ring tightly fitting around the cylinder was set about 2 mm away from the slip surface (Supplementary Fig. S1c).

The experimental conditions are summarized in Supplementary Table S1. In the experiments, the samples were immersed in deionized water of about 200 ml. We used a gas booster to pressurize pure nitrogen and a precision pressure-reducing regulator to control the nitrogen pressure at the upper part of the pressure vessel, through which the desired pore water pressure inside the vessel was obtained (Supplementary Fig. S1b). For a given set point of the pressure-reducing regulator, if the downstream pressure increases due to temperature rise or volume change, the poppet valve inside the regulator would be closed, making a closed system of the vessel during the experiments. It is worth noting that any changes in local pressure could be significantly buffered due to the high compressibility of nitrogen and the relatively large volume of water in the vessel. For the sample configuration we used, the pore pressure could impose a downward force ($F_{pp}$) that reduces the net axial loading on the slip surface. In the data processing, we used the recorded data of $P_p$ and axial force from the air actuator ($F_a$), and the calibrated relation[25] between $F_{pp}$ and $P_p$ to determine $\sigma_{n\_eff}$ (= ($F_a - F_{pp}$)/$A$ = [$F_a$ − (177.9*$P_p$ [MPa] + 90.5)]/$A$, where $A$ is the area of the slip surface). The changes in bulk $P_p$ only brought about small deviations (<0.3 MPa) from the initial $\sigma_{n\_eff}$ during the experiments.

The sheathed thermocouple of 0.5 mm in diameter was used to monitor the temperature evolution in selected experiments. The thermocouple hole of 2 mm in diameter was drilled through the stationary rock cylinder, and the thermocouple was fixed in the hole by using the high-temperature waterproof adhesive, with its tip end exposing on the slip surface (Supplementary Fig. S1b, c).

In each experiment, before shearing the dolerite samples at the equivalent slip rate[24] of 2.0 m/s, we preslid the samples for two revolutions (~166 mm in equivalent displacement) under the slip rate of 5 mm/s and the normal stress same as that in the main test. This adjusts the alignment of rock cylinders and helps to get better quality experimental data.

### Numerical modeling of thermal pressurization

We used dolerite sample in our experiments, so the thermochemical effects such as thermal decomposition can be ignored in TP modeling. To avoid complexities associated with phase transition of water (e.g., latent heat of vaporization and discontinuous changes in physical properties of water), we did TP modeling for the selected experiments under pore pressure of 25 MPa. If we also ignore the inelastic porosity changes, according to the energy and fluid mass conservation, the governing equations of TP are[9]:

$$\rho c \frac{\partial T}{\partial t} = \nabla \cdot (\lambda \nabla T) + Q_{\text{fric}} \qquad (1)$$

$$S_s \frac{\partial P}{\partial t} = \nabla \cdot \left( \frac{k}{\eta} \nabla P \right) + \varphi \left( \alpha_f - \alpha_m \right) \frac{\partial T}{\partial t} \qquad (2)$$

where $T$ is temperature, $P$ is pressure, $t$ is time, $\rho$ is density, $c$ is specific heat capacity, $\lambda$ is thermal conductivity, $Q_{\text{fric}}$ is heat generation rate by frictional heating, $S_s$ is specific storage, $k$ is permeability, $\eta$ is dynamic viscosity of fluid, $\alpha_f$ and $\alpha_m$ are thermal expansivities of fluid and mineral matrix, and $\Phi$ is porosity. The specific storage $S_s$ is expressed as[10]:

$$S_s = \beta_b + \varphi \beta_f - (1 + \varphi) \beta_m \qquad (3)$$

where $\beta_f$, $\beta_m$, and $\beta_b$ are the compressibilities of pore fluid, individual mineral grains and bulk sample, respectively.

To estimate $Q_{\text{fric}}$, we have to assume the shear-zone width, the velocity distribution in the shear zone and the intrinsic friction coefficient properly. Here we simply avoid the uncertainties in the assumption of these parameters by ignoring the $Q_{\text{fric}}$ term and setting instead a time-dependent temperature boundary $T(r, z=0, t)$ extrapolated from the measured temperature (here $r$ is the distance from the center and $z$ is the position along $z$ axis). To be specific, the $T(r, z=0, t)$ is assumed to be directly proportional to $r$ at $r \geq 1$ mm, and spatially constant at $r < 1$ mm (= $T(r=1, z=0, t)$; for the purpose of allowing the temperature to evolve reasonably in the center), with the $T(r = 15$ mm, $z = 0, t)$ given by thermocouple. Such treatment is actually equivalent to assuming a boundary heat source. Considering the TP weakening, the shear torque ($Tor$) can be expressed as:

$$\text{Tor} = \int_0^{r_0} \mu_i [\sigma_n - P_{\text{ave}}(r)] 2\pi r^2 \mathrm{d}r \qquad (4)$$

where $\mu_i$ is the intrinsic friction coefficient, $r_0$ is the radius of the rock cylinder, $\sigma_n$ is the applied normal stress and $P_{\text{ave}}(r)$ is the averaged pore pressure along $z$ axis. Here $\mu_i$ can be extrapolated from the slip-weakening data obtained in the room-humidity experiment (Fig. 2a), and $P_{\text{ave}}(r)$ can be written as, $P_{\text{ave}}(r) = \int_{-W/2}^{W/2} P(r,z) \mathrm{d}z / W$, where $W$ is the thickness of the shear zone. Then the apparent friction coefficient ($\mu_{\text{ap}}$) can be defined such that the shear torque is

$$\text{Tor} = \int_0^{r_0} \mu_{ap} \sigma_n 2\pi r^2 \mathrm{d}r \qquad (5)$$

The Eqs. (4) and (5) yield

$$\mu_{ap} = \frac{\int_0^{r_0} \mu_i[\sigma_n - P_{ave}(r)]2\pi r^2 dr}{\int_0^{r_0} \sigma_n 2\pi r^2 dr} \tag{6}$$

The wear-induced sealing effects were evaluated by considering the heterogeneity of hydraulic properties in the shear zone. We built 2D axisymmetric finite element models using Comsol Multiphysics. As is shown in Supplementary Fig. S6a, b, a 50 μm thick shear zone is sandwiched between a pair of dolerite cylinders with a radius of 20 mm. This is a proper model assumption because (1) a thin gouge layer may form soon after the sliding commences due to comminution and wear on the initial bare surface; and (2) the shear textures shown in Fig. 3a–c and the final thickness of molten layer (~100 μm) suggest that the shear-zone thickness in the early stages of the experiments should be a few tens of microns. Why we believe the hydraulic properties in the shear zone must be highly heterogeneous lies in the facts that the degree of shear deformation varies a lot with location (Fig. 3a–e). We could expect that the high-sheared zone such as the slickensided area should have lower permeability and porosity than the weakly-deformed zone. We tried two kinds of geometric models in the TP modeling, with either one or three ring-shaped high-sheared zones (acting as sealing zones) within the less-deformed bulk shear zone (Supplementary Fig. S6a, b). The assumption of the ring-shaped sealing zone(s) is based on the microstructural observation that local melting and severe wear occur along the ring-shaped slickensides (Fig. 3a–e and Supplementary Fig. S5a). As the measured bulk pore pressure rise ranged from ~0.15–1.6 MPa during the TSW, the pressure at the periphery of the shear zone was simply assumed to be constant at 25 MPa in the modeling.

The thermal, hydraulic, and other physical properties involved in the modeling are summarized in Supplementary Table S2. Fluid transport properties are decisive in TP process. Here the selection of permeability and porosity roughly follows that reported in Oohashi, et al.[26], where the permeability of granular and clay-rich gouges before and after rotary-shear friction experiments were measured. The permeability range of $10^{-14}$ to $10^{-20}$ m$^2$ they reported provides possible permeability bounds of the shear and sealing zones in our modeling. Considering cracking and fracturing may significantly increase permeability, we extended the upper permeability bound of bulk shear zone to be $10^{-8}$ m$^2$. The modeling using various combinations of permeability and porosity of bulk shear zone and sealing zone allows us to evaluate (1) the TP weakening in the case of uniform permeability and porosity in the entire shear zone (Group 1 parameters in Supplementary Fig. S6d; equivalent to the case of no sealing zone(s); modeling results shown in Fig. 4a and Supplementary Fig. S7a); (2) the sealing effects on the TP weakening within both permeable and impermeable bulk shear zones (Group 2 parameters in Supplementary Fig. S6d; Fig. 4b and Supplementary Fig. S7b); (3) how the amount of water within shear zone may affect the sealing-enhanced TP weakening (Group 3 parameters in Supplementary Fig. S6d; Fig. 4c and Supplementary Fig. S7c); (4) how the permeability of sealing zone may influence the TP weakening with the sealing effects being considered (Group 4 parameters in Supplementary Fig. S6d; Fig. 4d and Supplementary Fig. S7d). The modeling with one (Fig. 4) and three (Supplementary Fig. S7) ring-shaped sealing zones yield more or less similar results.

It is noteworthy that the real transient TP processes in our experiments were far more complicated than what we could consider in the modeling. For instance, the geometry of the active sealing zone may evolve rapidly during slip, and the sealing zone may have limited sealing capability, and other fluid flow processes beyond Darcy flow (e.g., fracture flow) may occur during the complicated deformation processes. Due to these limitations, we take the modeling results as first-order assessments of the TP weakening in the experiments.

## Data availability
All experimental data and two examples of the numerical models are available in Zenodo under https://doi.org/10.5281/zenodo.7438977.

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

## Acknowledgements

This work was supported by the National Natural Science Foundation of China Grants 41774191 and 42111530030 to L.Y., and Grant U1839211 to S.M., the Institute of Geology, China Earthquake Administration Grant IGCEA2107 to L.Y., and ERC Consolidator Grant 614705 NOFEAR to G.D.T. We thank Toshihiko Shimamoto for designing and testing the experimental machine, and Yu Wang for her great help in microstructural observation. L.Y. would like to thank Nadia Lapusta, Jianye Chen, Wei Feng, Lei Zhang, Qingbao Duan, Miao Zhang and Rodrigo Gomila for useful discussions.

## Author contributions

L.Y. and S.M. conceived the study. L.Y. performed friction experiments, microstructural analysis, and numerical modeling. L.Y. and G.D.T. prepared the manuscript and figures. All the authors contributed to the interpretation of all results.

## Competing interests

The authors declare no competing interests.
