## [Peer Review File · Nature Communications]

REVIEWER COMMENTS

Reviewer #1 (Remarks to the Author):

This paper reports a well-conceived and conducted experimental study of frictional weakening and restrengthening during a single, large, slip event on a bare rock surface using a rotary shear apparatus. The experiments are extremely well done, monitored, imaged, and analyzed. The interpretations of mechanical behavior in terms of initial weakening by flash-heating, followed by restrengthening with onset of melting, and then a transient dramatic weakening (interpreted as thermal fluid pressurization), and finally a general moderate restrengthen from development of a continuous melt layer to achieve viscous flow, are all well described. These phenomena have been previously documented, except for the new observation of thermal fluid pressurization that produces sudden, temporary weakening to near-zero frictional strength. The authors interpretation of wear product sealing to prevent fluids from escaping the sliding surface is plausible, and is supported by the documentation of enhanced weakening effect when machined cavities along the interface are added to provide greater fluid volume. In my opinion, a key component of the study are the images of the sliding surface (Fig 3 a-c) taken from an experiment (LHV1103) in which was slipped only to just after the approximate displacement of the transient sharp weakening. The numerical modeling also explores the effect of porosity and permeability in the experiments to support plausibility of the interpretation, but as noted by the authors, the main interpretation of wear product sealing to achieve fluid pressurization is not evaluated by the model.

This paper is a solid contribution, and presents experimental observations and interpretations of dynamic weakening mechanisms in seismic faulting in the crust, which is of great interest to the seismology, rock mechanics, structure and tectonics communities investigating earthquake physics. The developed understanding of earthquake rupture processes is incorporated into dynamic rupture modeling that allows evaluation and prediction of earthquake effects and seismic hazard. Although this paper follows many other studies using rotary shear friction testing, duplicating earlier results associated with flash heating and melt generation in large displacement sliding tests, the observation of thermal fluid pressurization in such experiments is a unique advance, and will likely have significant impact.

The paper is nicely prepared, and in my view needs only minor revision for publication. Below are a few points that the authors should address.

The documentation of the mechanical data is fairly complete, but is missing experiment LHV1103, why? Experiment LHV1103 was terminated at 4.5 m of slip, presumably immediately after the sharp transient weakening event, and thus illustrating the surface characteristics tied to that weakening.

As a reader, I want to see the nature of the weakening in that particular experiment and how it compares mechanically with LHV968.

Figure 4 is a nice illustration and description of the inferred changes in slip processes tied to weakening and restrengthening phases seen in the tests. It obviously applies to the experiments directly in that there is an existing sliding surface that is activated producing a very thin wear product layer. I'm curious if in the experiment there actually is microcracking of the rock as depicted in the illustrations. I believe this figure would be improved with labels of approximate magnitudes on the axes, in particular the axes of time and displacement.

I am less enthusiastic about Figure 1b. It is illustrated similar to Fig. 4, but is intended to apply to a large slip event on an established earthquake fault, producing an earthquake of magnitude ~ 6 or so (shown in Fig. 1a) with displacement sufficient to produce melt. In this case such a slip event would not be a rupture along a pristine surface as in the experiment, but along a localized slip zone of an established fault, presumably bounded by a broader zone of extensive damage. So, one problem with Figure 1b is that it shows 1 mm-size grain along the slip surface rather than fine grained gouge or cataclasite. As shown, Fig. 1b implies that the representation of the stages of slip in the experiment can be applied directly to a natural established fault hosting a large displacement earthquake. This deserves some mention in the text if Fig. 1b is retained.

Fig. 1b is modified from Ref. 21, and thus may be applicable to a rupture in intact rock in the deep crust. (Although I have some difficulty understanding the rupture tip described in Ref. 21 that achieves 1.3 m offset within a few meters from the tip, in a single exposure with effectively no evidence of finite strain in the host rock that would be necessary to accommodate the implied slip gradients.). Regardless, the other significant issue with Fig 1b is not showing representative magnitudes of EQ slip, position and time on the horizontal axis from the tip (axis currently labeled "Time since the rupture tip passed" without any values). Also, having a horizontal scale bar of 1 mm is even more confusing. I suppose if the 1 mm scale bar is intended for the vertical direction (rotated 90°) it would be better, but still misleading if the horizontal length scale (orders of magnitude larger) is not included. In my understanding, if one assumes melt layer generation in nature would require slip of 0.1 m or more to produce injections, and for a slip rate of 1 m/s, then the time of melt layer development (since passage of rupture front) would be a significant fraction of a second. Thus, the horizontal distance of the figure is a significant fraction of a km given the rupture front speed. I think it is critical to point this out with labels on the figure for the reader.

Fixing Fig. 1b might provide a good opportunity to discuss in the conclusion section the application of the findings of the paper to natural EQ faulting in the brittle upper crust.

Reviewer #2 (Remarks to the Author):

The manuscript entitled “Coseismic fault sealing and fluid pressurization during earthquakes” describes transient sharp weakening of water-saturated fault during high-velocity friction tests, and results of textural observations and numerical modeling. The authors concluded that the behavior is attributable to wear-induced sealing and thermal pressurization during the shearing. Dynamic weakening process of seismic faults is crucial to understand the generation and propagation of earthquake rupture. However, due to the technical difficulties, there is still a lack of experimental demonstration. This study, done with a newly designed experimental configuration and model analysis, may shed light on new understanding of coseismic fault process. The manuscript is basically well written, straightforward, and their results seem sound. However, I found several descriptions are inappropriate or unclear. Hence I recommend accepting it with minor revisions.

Comments

(1) Lines 45-52; Here, the authors explain the coseismic physico-chemical processes within the slipping zone from (1) to (4), referring to Figure 1. I would suggest adding the number (1) to (4) to corresponding process shown in the figure so that the reader can easily imagine what you mean (e.g., (1) Asymmetric crack formation). Also, the authors mentioned “(1) fracturing and fragmentation occur due to the increased stress level near the rupture tip (asymmetric cracks may form)” but the text “Rock fragmentation” is placed between “Rock fragmentation” and “Intense wear” in the figure, and seems to occur at the second or third stage of whole processes.

(2) Lines 73-74; “The friction coefficient (μ) increases to a first peak of 0.52 and drops quickly to 0.35 (red line). Then μ increases to a higher peak of 0.71,”

I would suggest the following change; “The friction coefficient (μ) rapidly increases to a first peak of 0.52 and drops quickly to 0.35 (red line). Then μ gradually increases to a higher second peak of 0.71,”

(3) Line 66; I would suggest adding the following description; “We used high-purity nitrogen gas and deionized water for pore fluids.”

(4) Figure 2; There is no temperature and/or axial displacement scales in Figure 2a and c. Although, we could refer them in b and d, it is a bit confusing and hardly recognized the value without using a ruler.

(5) Line 116; “glassy-like rods”

The authors use the term “glassy-like rods” to describe a melt layer (or patch) under SEM observations. I have two comments here. (1) I prefer use “glassy rods” or “glass-like rods” instead of “glassy-like rods”. (2) How do you find it to “glassy” under SEM observations? I think it is not suitable to use “glassy” because it could only be distinguished under optical light microscopes (or X-ray/electron diffraction). If you also conducted surface/thin-section observations for those specimens under optical light microscopes, please mention it in the text.

(6) Figure 3; There is three SEM photographs (a, b and c) taken from different locations of identical slip surface. Can you provide the information about the location of each photographs by adding top-view image of slip surface? I think it is necessary in evaluating validity of model geometry shown in Extended Data Figure 5.

(7) Line 156; “relatively low permeability” must be changed to “low permeability”.

(8) Lines 163-164; “observed re-strengthening stage (Figs. 2b–d and Extended Data Fig. 2c) may be partly a consequence of the heterogeneous fluid transport properties.”

I could not understand what you meant. Could you be more specific?

(9) Lines 173-174; “These two processes may also explain the observed changes in the shortening rate.”

Again, I could not understand what you meant here. Could you be more specific?

(10) Conclusion; I think it is better mentioning about the low effective normal stress observed in natural seismogenic fault zones (e.g., Audet et al., 2009; Peacock et al., 2011) to increase the applicability and impact of this result.

(11) Extended Data Figure 2; I could not understand why the experiment with N₂ gas shows TSW as same as the one with water, despite there must be a large difference in thermal properties (e.g., thermal expansivity) and viscosity. Some explanation or model calculation must be needed.

Reference:

Audet, P., Bostock, M. G., Christensen, N. I., and Peacock, S. M.: Seismic evidence for overpressured subducted oceanic crust and megathrust fault sealing, *Nature*, 457, 76–78, <https://doi.org/10.1038/nature07650>, 2009.

Simon M. Peacock, Nikolas I. Christensen, Michael G. Bostock, Pascal Audet; High pore pressures and porosity at 35 km depth in the Cascadia subduction zone. *Geology* 2011;; 39 (5): 471–474. doi: <https://doi.org/10.1130/G31649.1>

Best wishes,

Kiyokazu Oohashi

Reviewer #3 (Remarks to the Author):

Review

The manuscript by Yao et al. reports the results of high-speed friction experiments on the dolerite samples sheared under the condition of fluid pressure applied. The experiment configurations are novel to simulate the earthquake slip at depths more realistically. The authors also introduced a new “pitted sample” configuration with some holes to trap some fluid at the slip interface. They observed very drastic and sharp weakening and concluded with the integration of microstructure observation and numerical modeling that it is due to local thermal pressurization due to the transient local shear zone developed due to wear and melt products during the shear. Their techniques are novel and essential for advancing our understanding of earthquake slips and processes. However, due to the complexity of the experiment configuration and complicated kinetics and deformation processes, it is very challenging to understand what is happening during the test. I have concerns about some descriptions and interpretations, primarily due to the lack of critical information. Below, I made comments separately for friction experiments, microstructure observation, and numerical simulations.

<Friction Experiments>

Looking at the friction coefficient and temperature evolution shown in Extended data figure 2, I am not sure what the effective stress means in this configuration. As it is said in Lines 55-56, the samples “initially have open boundaries for fluid flow.” Clearly, confining pressure is equal to pore fluid pressure in this sample configuration.

I believe N₂ pressure is kept constant at the upper part of the pressure vessel during the test to keep the water or N₂ P_p constant, correct? It is said in Lines 231-232 that any changes in local pressure (I'm not 100% sure what "local pressure" means) could be significantly buffered due to the high compressibility of nitrogen and the relatively large volume of water in the vessel. However, the authors observed an increase in P_p on the transducer located at the lower part of the pressure vessel. Does that mean the pore pressure pressurization system can't catch up with rapid sample responses such as P_p change (and sample dilation/compaction) within the shear zone during the high-speed shear? Or do you have a closed system ("undrained") that does not intend to control the pore pressure constant during the test?

Also, pore pressure measurement on the pressure transducer was not shown for the entire duration of the experiments (Extended Data Figure 2; c or d). Why?

Related to the above question, how is the experiment's friction coefficient calculated? I believe you divide shear stress, which is calculated from the measured torque, by effective normal stress, but do you assume the effective normal stress is constant during the shear? The effective normal stress changes during the shear as the pore pressure changes. I understand it is probably impossible to quantify the effective normal stress within the shear zone because of the unknown pore pressure there; however, the measured pore pressure (called bulk pore pressure in the manuscript) also changes. Do you account for the bulk pore pressure change to calculate effective normal stress? The effect is probably small, but it would be good to know how the reported values are calculated.

For the tests with pore fluid of water, I can see relatively quick recovery followed by a drastic drop in frictional coefficient at a displacement of ~9 m and ~12 m (Extended data Figure 2). The caption says, "the large fluctuation in friction during the re-strengthening stage in the cases of pore water pressure (LHV968 and LHV969) is probably caused by the complicated fracturing, wear-induced sealing, and bulk melting processes in the presence of water." The response in both tests is very similar; thus, I wonder if some systematic mechanism causes the fluctuation. Also, the temperature evolution in LHV969 and LHV965 becomes similar after ~15 m slip. Do you think that's the time bulk melting occurs? The temperature of LHV 968 after ~15 m is not shown in Figure. Is that because the thermocouple was damaged? Some explanation would be great to have. By the way, it is hard to distinguish the temperature data shown on panel d of Extended Data Figure 2. Could you use the solid line with different colors shown on panel c?

Extended data figure 2d: In the figure caption, it is mentioned that "the overall significantly lower temperature in the case of 10 MPa water should be attributed to the energy sink associated with the water phase change at the boiling point, which is 311 °C at P_p = 10 MPa (much lower than 591 °C)". I don't fully understand the argument if you are talking about the difference between LHV969 (P_p = 10 MPa with water) and LHV 968 (P_p = 25 MPa with water). The difference in temperature between

LHV969 ($P_p = 10$ MPa with water) and LHV968 ($P_p = 25$ MPa with water) is likely attributed to the peak frictional coefficient observed at the beginning of the shear.

Transient sharp weakening (TSW) accompanies by an abrupt increase in the rate of bulk pore pressure rise, and I noticed that P_p raise is relatively large; for example, in LHV968, the bulk pore pressure rise reaches ~ 1.7 MPa, which is about 7 % of the pore pressure (25 MPa). Have you calculated how much volume change is required to cause such a rise in a short period? As I mentioned earlier, I'm unsure if the pore pressure is controlled at constant pressure at the upper side of the pressure vessel.

The authors also described a possible correlation between TSW and axial shortening in Lines 106 and 110. However, no interpretation was provided on it. What causes shortening? A question is whether wear/melt products of dolerite are kept in place during/after the shear or if some are floating or deposited somewhere in the pressure vessel. Because the sample configuration cannot confine the sample, losing some materials leads to shortening.

Temperature data of the experiments at effective normal stress = 6 or 10 MPa were not presented. Are they not measured? Just not presented? These are test conditions where you compared pitted samples vs. flat samples. I am not sure how the pitted samples contributed to the pore pressure evolution or TSW because TSW is observed on the flat samples at effective normal stress = 3 MPa.

For "pitted sample" tests, do both stationary and rotary blocks have pits or only one block?

Where are the pits located? In Extended data figure 1d, I can see one pit at the center and the other 8 holes, but are they located at a distance from the center? If so, how far are they?

Please clarify.

The location (distance from the center) of the thermocouple in the experiment set was not mentioned clearly. I believe it is located 15 mm from the center based on the information given for the numerical simulation. Please clarify.

<Microstructure observations>

The microstructures were presented for the test LHV1103, with a total slip of 4.5 m in Figure 3 a-c and for LHV968, with a slip of 20 m in Extended data figure 4. No information on the location of each image. There is some inset for Figures 3b and 3c, but each image's location and direction are unclear (e.g., distance from the center; slip parallel or slip perpendicular). Also, the mechanical data of this

experiment (LHV1103) was not presented. I understand the test conditions of normal stress, P_p , and slip rate are the same as LHV968, but I'm not sure the mechanical response is the same. Please add some description in the text or present the mechanical data of LHV 1103.

What is the expected melting temperature of your dolerite sample?

<FEM Modeling>

It is unclear why you have three ring seal zones vs. one seal zone for the modeling setup. I don't see any discussion on the difference between single and three seal zones. I don't also know what motivated authors to set up three ring seal zones. Are they based on some microstructure observations?

What is the time = 0 of the modeling? At the beginning of TSW?

Figure 3:

In panel d, are the data for Dry, and $k_{bsz}=k_{seal}=10^{-10}$ and 10^{-11} shown here? I don't see them, but they may be behind the result of $k_{bsz}=k_{seal}=10^{-12}$.

In pane e, I don't see the data for $k_{bsz} = 10^{-8}$ to 10^{-12} ...

How about the boundary conditions at the end of the samples? The sample is immersed in the fluid (water) in the pressure vessel. Do you have a boundary condition of no flow or constant pressure or anything else at the periphery of the shear zone?

The description of the heat source setting (Lines 258-262) was unclear. The authors set up a localized line heat source at the center of the shear zone constrained by the time-dependent temperature. The temperature is a function of the sample radius (distance from the sample center), $T(r,t)$ is proportional to r , and $T(r = 15 \text{ mm}, t)$ is constrained by the thermocouple measurements during the friction experiments. If you assume $T(r,t)$ is proportional to r , does that mean that $T(r = 0, t)$ is fixed (at room temperature?) and stays constant during the shear? If that's the case, I don't think this is an appropriate setting. At the center of the sample, heat should not be generated because the shear velocity is zero, but that is not the same as constant temperature. So, I'm concerned that the heat source is constrained by the temperature. Please clarify this.

<Minor comments>

Lin 95: "Extended Data Fig s. 1a-c" should be "Extended Data Fig s. 2a-c"?

Line 111: "weld" should be "welded"

Reviewers' comments and our responses

In the following response letter, we listed the reviewers' comments in black and wrote our responses in blue following each comment. The Manuscript and the Supplemental Material have been modified in response to the comments. The changes we made were highlighted in blue in the annotated manuscript. The line numbers we mentioned below refer to those in the annotated manuscript.

Reviewer #1 (Remarks to the Author):

(1) This paper reports a well-conceived and conducted experimental study of frictional weakening and restrengthening during a single, large, slip event on a bare rock surface using a rotary shear apparatus. The experiments are extremely well done, monitored, imaged, and analyzed. The interpretations of mechanical behavior in terms of initial weakening by flash-heating, followed by restrengthening with onset of melting, and then a transient dramatic weakening (interpreted as thermal fluid pressurization), and finally a general moderate restrengthen from development of a continuous melt layer to achieve viscous flow, are all well described. These phenomena have been previously documented, except for the new observation of thermal fluid pressurization that produces sudden, temporary weakening to near-zero frictional strength. The authors interpretation of wear product sealing to prevent fluids from escaping the sliding surface is plausible, and is supported by the documentation of enhanced weakening effect when machined cavities along the interface are added to provide greater fluid volume. In my opinion, a key component of the study are the images of the sliding surface (Fig 3 a-c) taken from an experiment (LHV1103) in which was slipped only to just after the approximate displacement of the transient sharp weakening. The numerical modeling also explores the effect of porosity and permeability in the experiments to support plausibility of the interpretation, but as noted by the authors, the main interpretation of wear product sealing to achieve fluid pressurization is not evaluated by the model.

This paper is a solid contribution, and presents experimental observations and interpretations of dynamic weakening mechanisms in seismic faulting in the crust, which is of great interest to the seismology, rock mechanics, structure and tectonics communities investigating earthquake physics. The developed understanding of earthquake rupture processes is incorporated into dynamic rupture modeling that allows evaluation and prediction of earthquake effects and seismic hazard. Although this paper follows many other studies using rotary shear friction testing, duplicating earlier results associated with flash heating and melt generation in large displacement sliding tests,

the observation of thermal fluid pressurization in such experiments is a unique advance, and will likely have significant impact.

The paper is nicely prepared, and in my view needs only minor revision for publication. Below are a few points that the authors should address.

[Our response] We sincerely thank the reviewer for the kind words, incisive comments and valuable suggestions. We will give our response to each comment below. Here we briefly respond to the reviewer's foregoing comment about the numerical modeling (at the end of the first paragraph).

As illustrated in Fig. 4 (now Fig. 5 in the revised manuscript) and stressed in the last paragraph of the Methods section, the physico-chemical processes involved in our experiments are far more complicated than what the controlling equations of our numerical modeling, i.e., thermal and hydraulic diffusion equations, could describe. Precise evaluations of the wear-induced sealing effects are hindered by a lack of information about (1) the geometry and (2) the sealing capability of the active sealing zone, and (3) their evolution with slip. Also, little is known about to what extent other fluid flow processes beyond Darcy flow (e.g., fracture flow) may occur during the experiments.

Despite the limitations, our modeling results provide evaluations of how thermal pressurization may operate if the sealing zones stably exist during the time period studied. From this perspective, we think the modeling results could act as first-order assessments of the wear-induced sealing processes in the experiments.

(2) The documentation of the mechanical data is fairly complete, but is missing experiment LHV1103, why? Experiment LHV1103 was terminated at 4.5 m of slip, presumably immediately after the sharp transient weakening event, and thus illustrating the surface characteristics tied to that weakening. As a reader, I want to see the nature of the weakening in that particular experiment and how it compares mechanically with LHV968.

[Our response] The friction evolution prior to the peak friction is complicated in the experiment LHV1103. As is shown in Fig. (a) below, the friction coefficient goes up and down in the first 2 m slip, then increases rapidly towards a peak at a displacement of 4.0 m, after which sharp weakening occurs before the experiment stops.

In rock-on-rock experiments (i.e., using a pair of rock cylinders) performed at high slip rates, *the initial contact status of the slip surfaces may vary from test to test due to the misalignment of the two rock cylinders, so does the evolution of pre-peak friction.* Such an effect is much stronger in rock-on-rock experiments than in gouge experiments as the gouge layers are compliant. As a consequence, *the reproducibility of the pre-peak friction data is often not good in rock-on-rock experiments, especially for solid rock cylinders* (Fig. b below). Here the scatter in the mechanical data regarding the initial evolution of the friction coefficient with slip displacement. Once this initial evolution is overcome, and friction decays towards the so-called "steady-state" conditions, especially in the absence of pressurized fluids, the friction evolution seems to be highly

reproducible (Fig. b below; see also Niemeijer et al., JGR, 2011). Moreover, the presence of two friction peaks and the scatter in the friction data during the initial acceleration and slip is quite common in solid cylindrical samples due to the complicated thermal evolution of the slip zone (see discussion in Fialko and Khazan, JGR 2005 and in Nielsen et al., JGR 2008).

By shifting the displacement at which the friction reaches the peak to 0 m for the tests LHV968 and LHV1103, we think the run LHV1103 was terminated at the onset or initial stage of the sharp transient weakening (TSW). For the purpose of constraining the physical process associated with the TSW, it is reasonable to observe the microstructures of the slip surfaces just prior to or at the beginning of the TSW.

In the revised manuscript, the above figure is presented as Supplementary Fig. S4. We also briefly describe what we discussed above in the caption of the figure.

(3) Figure 4 is a nice illustration and description of the inferred changes in slip processes tied to weakening and restrengthening phases seen in the tests. It obviously applies to the experiments directly in that there is an existing sliding surface that is activated producing a very thin wear product layer. I'm curious if in the experiment there actually is microcracking of the rock as depicted in the illustrations. I believe this figure would be improved with labels of approximate magnitudes on the axes, in particular the axes of time and displacement.

[Our response] To confirm the microcracking of the rock in the experiment, we show the following SEM image of the sample recovered from the run LHV968 (back-scattered electron image). The bright-gray-colored region in the center is the molten layer consisting of melts and clasts. In the surrounding wall rocks, microcracks are widely distributed, suggesting the occurrence of microcracking associated with the intense shear deformation and sharp temperature rise during the experiment.

The plot of friction coefficient/temperature/pore pressure/shortening versus time/displacement in previous Fig. 4 (now Fig. 5 in the revised manuscript) is actually a schematic representation prepared based on the data shown in Fig. 2c. Following the reviewer's suggestion and considering how to make the diagram clear and simple, we simply added a scale bar to indicate the magnitudes of time and displacement.

(4) I am less enthusiastic about Figure 1b. It is illustrated similar to Fig. 4, but is intended to apply to a large slip event on an established earthquake fault, producing an earthquake of magnitude ~6 or so (shown in Fig. 1a) with displacement sufficient to produce melt. In this case such a slip event would not be a rupture along a pristine surface as in the experiment, but along a localized slip zone of an established fault, presumably bounded by a broader zone of extensive damage. So, one problem with Figure 1b is that it shows 1 mm-size grain along the slip surface rather than fine grained gouge or cataclasite. As shown, Fig. 1b implies that the representation of the stages of slip in the experiment can be applied directly to a natural established fault hosting a large displacement earthquake. This deserves some mention in the text if Fig. 1b is retained.

[Our response] We agree that natural fault zones hosting moderate to large earthquakes must be very different from experimental faults in terms of internal structures. At the depth of ~ 5 km, as illustrated in Fig. 1b, the principal slip zone (PSZ) accommodating the seismic slip should be typically developed within cataclasites (Sibson, 1977). However, whether the cataclasites surrounding the PSZ are coarse- or fine-grained may depend on the maturity, the lithology, the mechanical properties and the geological setting of the fault zone, etc., as suggested by field observations on exhumed natural faults worldwide. For instance, the exhumed Punchbowl fault of the San Andreas fault system contains a continuous prominent slip surface developed within a ~0.2 m thick ultracataclasite layer at a few exposures near LA (Chester and Chester, 1998, Tectonophysics). The rocks surrounding the PSZ in such a case are indeed fine-grained (mostly < 10 μm). Yet in some of the exhumed pseudotachylyte-bearing faults, like the

Gole Larghe fault in Italian Southern Alps and the southern section of the Longmenshan fault that hosts the 2008 Mw 7.9 Wenchuan earthquake (exposures near Hongkou), the pseudotachylyte veins (fossil PSZs) are located within medium-grained (~mm) cataclasites or protocataclasites (see Fig. 2 of Di Toro and Pennacchioni, 2004, JSG; Fig. 4 of Wang et al., 2014, Tectonophysics). Actually, during post-seismic healing and sealing, micrometer-in-size grains may be cemented into larger clasts (> ~mm in size). In short, we think a few μm to mm is a reasonable range of mineral grains surrounding the PSZ within cataclasites at the middle crust.

In the revised manuscript, for the purpose of a clear presentation, we don't make changes to Fig. 1b. But we address the scale issue by stating that Fig. 1b is drawn not to scale in the caption (see details below).

(5) Fig. 1b is modified from Ref. 21, and thus may be applicable to a rupture in intact rock in the deep crust. (Although I have some difficulty understanding the rupture tip described in Ref. 21 that achieves 1.3 m offset within a few meters from the tip, in a single exposure with effectively no evidence of finite strain in the host rock that would be necessary to accommodate the implied slip gradients.). Regardless, the other significant issue with Fig 1b is not showing representative magnitudes of EQ slip, position and time on the horizontal axis from the tip (axis currently labeled "Time since the rupture tip passed" without any values). Also, having a horizontal scale bar of 1 mm is even more confusing. I suppose if the 1 mm scale bar is intended for the vertical direction (rotated 90°) it would be better, but still misleading if the horizontal length scale (orders of magnitude larger) is not included. In my understanding, if one assumes melt layer generation in nature would require slip of 0.1 m or more to produce injections, and for a slip rate of 1 m/s, then the time of melt layer development (since passage of rupture front) would be a significant fraction of a second. Thus, the horizontal distance of the figure is a significant fraction of a km given the rupture front speed. I think it is critical to point this out with labels on the figure for the reader.

Fixing Fig. 1b might provide a good opportunity to discuss in the conclusion section the application of the findings of the paper to natural EQ faulting in the brittle upper crust.

[Our response] *We appreciate the reviewer's comments on the timing and scale issues relevant to the process shown in Fig. 1b, which we totally agree about.* As the reviewer pointed out, a shear displacement of ~0.1–1 m is probably required for the formation of a molten layer, which may take about 0.1–1 s for the slip velocity of ~1 m/s. We thus add the value of 0.1–1 s on the horizontal axis to show the timing of the process. Given the rupture speed of ~3 km/s, the location where bulk melting occurs may be at least a few hundred to thousand meters away from the rupture tip (so is the horizontal distance of Fig. 1b). Moreover, the thickness of fault-hosted pseudotachylyte is typically a few millimeters to centimeters (e.g., Sibson and Toy, 2006; Di Toro and Pennacchioni, 2004, JSG), and the grain size of the ultracataclasite/cataclasite/protocataclasite surrounding the principal slip zone (PSZ) may be reasonably within the range of ~10 μm –1 mm (as discussed above in response to the previous comment).

For the purpose of a clear presentation, we think it's better to draw Fig. 1b not to scale, i.e., showing mineral grains and the PSZ out of proportion to the dimension of the slipping region of the fault. The following sentences have been added in the caption of Fig. 1.

“Fig. 1b is drawn not to scale: the thickness of the molten layer, the size of the surrounding mineral grains, and the length of the slipping fault patch are a few mm to cm, a few μm to mm, and a few hundred to thousand m, respectively.” (Lines 64 to 66)

Reviewer #2 (Dr. Oohashi, Remarks to the Author):

The manuscript entitled “Coseismic fault sealing and fluid pressurization during earthquakes” describes transient sharp weakening of water-saturated fault during high-velocity friction tests, and results of textural observations and numerical modeling. The authors concluded that the behavior is attributable to wear-induced sealing and thermal pressurization during the shearing. Dynamic weakening process of seismic faults is crucial to understand the generation and propagation of earthquake rupture. However, due to the technical difficulties, there is still a lack of experimental demonstration. This study, done with a newly designed experimental configuration and model analysis, may shed light on new understanding of coseismic fault process. The manuscript is basically well written, straightforward, and their results seem sound. However, I found several descriptions are inappropriate or unclear. Hence I recommend accepting it with minor revisions.

[Our response] We sincerely thank Dr. Oohashi for the careful review, and the constructive comments and suggestions. We respond to the comments as follows.

Comments

(1) Lines 45-52; Here, the authors explain the coseismic physico-chemical processes within the slipping zone from (1) to (4), referring to Figure 1. I would suggest adding the number (1) to (4) to corresponding process shown in the figure so that the reader can easily imagine what you mean (e.g., (1) Asymmetric crack formation). Also, the authors mentioned “(1) fracturing and fragmentation occur due to the increased stress level near the rupture tip (asymmetric cracks may form)” but the text “Rock fragmentation” is placed between “Rock fragmentation” and “Intense wear” in the figure, and seems to occur at the second or third stage of whole processes.

[Our response] Thanks for the suggestion. Although the color of the text in Fig. 1b already denote the chronological order (dark green: $t = 0$ and dark orange: $t =$ ending time of the slip pulse; see color bar), we agree that numbering these processes may make the main text and Fig. 1b easier to follow. We have added superscript numbers to the relevant text in Fig. b as suggested.

The reviewer also commented that there seems to exist an inconsistency between the main text and Fig. 1b regarding the timing of “rock fragmentation”. In our opinion, rock fragmentation may occur during most time of the slip pulse, due to both stress perturbation during rupture propagation and subsequent mechanical deformation associated with fault slip. In the revised Fig. 1b, we have used the superscript numbers “1-3)” and the color gradient (transition from chartreuse to dark green) to indicate that rock fragmentation may occur from the 1st to 3rd stages of the whole processes. Also, we have added a few words in lines 48–49 and rearranged the locations of the text “rock fragmentation” and “flash weakening” in Fig. 1b.

(2) Lines 73-74; “The friction coefficient (μ) increases to a first peak of 0.52 and drops quickly to 0.35 (red line). Then μ increases to a higher peak of 0.71,”

I would suggest the following change; “The friction coefficient (μ) rapidly increases to a first peak of 0.52 and drops quickly to 0.35 (red line). Then μ gradually increases to a higher second peak of 0.71,”

[Our response] We have revised the sentence as suggested. (lines 79–80)

(3) Line 66; I would suggest adding the following description; “We used high-purity nitrogen gas and deionized water for pore fluids.”

[Our response] We have added the following sentence as suggested:

“Deionized water was used as pore fluid in most of the experiments (except for one run in which high-purity nitrogen filled the entire vessel), although the pore pressure was exerted by the pressurized nitrogen at the upper part of the vessel in all the experiments.”

(lines 70–73)

(4) Figure 2; There is no temperature and/or axial displacement scales in Figure 2a and c. Although, we could refer them in b and d, it is a bit confusing and hardly recognized the value without using a ruler.

[Our response] We have added axes, scales and ticks for temperature/pore pressure and axial displacement in Figure 2a and c as suggested.

(5) Line 116; “glassy-like rods”

The authors use the term “glassy-like rods” to describe a melt layer (or patch) under SEM observations. I have two comments here. (1) I prefer use “glassy rods” or “glass-like rods” instead of “glassy-like rods”. (2) How do you find it to “glassy” under SEM observations? I think it is not suitable to use “glassy” because it could only be distinguished under optical light microscopes (or X-ray/electron diffraction). If you also conducted surface/thin-section observations for those specimens under optical light microscopes, please mention it in the text.

[Our response] We agree that we should use the term “glassy rods” or “glass-like rods” rather than “glassy-like rods”.

The grain size of these rods is very small, < micrometer diameter, and their volume is very limited, so it is not possible to use the optical microscope or the XRD diffraction (maybe XRD in the synchrotron beam). Perhaps we should use FIB-TEM or high-resolution micro-Raman spectroscopy to confirm the presence of glass/melts. However, *the presented SEM images show textures, including rods/filaments interpreted as due to the stretching of melt drops (Figs. 3b), and glass-like surface coatings (previous Figs. 3c; now Fig. 3d in the revised manuscript)*. Similar, if not identical, microstructures were produced in previous studies and interpreted as the result of frictional melting associated with the first mm-cm of slip (e.g., Lockner et al., 2018, doi: 10.1002/9781119156895.ch6; Violay et al., Geology, 2014):

Ropy-like structures interpreted as glass features covering the slip surface in experiments performed on gabbro. The slip distance of this experiment was 5 mm and the max velocity was 0.23 m/s (Violay et al., Geology, 2014).

The formation and the dimensions of these drops of melts (and filaments when melt solidifies) at slip initiation is consistent with theoretical models of temperature increase in the slipping zone due to flash heating & melting for gabbro-diorite rocks and the transition from flash heating & melting to bulk melting and melt lubrication regimes (Cornelio et al., JGR, 2022).

(6) Figure 3; There is three SEM photographs (a, b and c) taken from different locations of identical slip surface. Can you provide the information about the location of each photographs by adding top-view image of slip surface? I think it is necessary in evaluating validity of model geometry shown in Extended Data Figure 5.

[Our response] Thanks for the constructive comments. *We have made substantial revisions to Fig. 3 and Supplementary Fig. S5 as suggested.* Now the locations of the

close-up SEM images (Figs. 3b–g) are indicated in the low-magnification images shown in Fig. 3a and Supplementary Figs. S5b–c (see revised description of Fig. 3 and Supplementary Fig. S5 in lines 118–132)

The merged secondary electron images in Fig. 3a and Supplementary Fig. S5a reveal *continuous ring-shaped slickensides, where melts and fine wear products seem to form locally*. We believe these portions of the slip surface may serve as sealing zones as we constructed in the numerical modeling. (lines 155–160; lines 331–333)

(7) Line 156; “relatively low permeability” must be changed to “low permeability”.

[Our response] We have deleted “relatively” as suggested.

(8) Lines 163-164; “observed re-strengthening stage (Figs. 2b–d and Extended Data Fig. 2c) may be partly a consequence of the heterogeneous fluid transport properties.” I could not understand what you meant. Could you be more specific?

[Our response] By performing numerical modeling with more combinations of relevant parameters when we revise the manuscript, we realize it is inappropriate to argue that the observed re-strengthening may be partly a consequence of the heterogeneous fluid transport properties. We have deleted the relevant description/discussion.

Actually, the modeling results with group-4 parameters shown in Supplementary Fig. 6 of the previous manuscript do not give very useful information. The k_{bsz} used there (k_{bsz} is fixed at 10^{-15} m^2 and k_{seal} is varied from 10^{-15} to 10^{-20} m^2) is not a reasonable setting (too low) for a permeable shear zone, which hinders us from evaluating the effects of k_{seal} on the sealing-enhanced TP in the modeling. In the revised manuscript, the group-4 parameters are set by fixing the k_{bsz} at 10^{-12} m^2 (assuming a highly permeable shear zone) and varying k_{seal} from 10^{-12} to 10^{-20} m^2 . Then the results give the range of k_{seal} required to activate the sealing-enhanced TP (Fig. 4d in the revised manuscript; lines 190–193), which is much more helpful in interpreting the experimental data.

(9) Lines 173-174; “These two processes may also explain the observed changes in the shortening rate.” Again, I could not understand what you meant here. Could you be more specific?

[Our response] The failure of the sealing and melt patches may result in the escape of pressurized water and thus an increase in σ_{n-eff} . The wear rate could be higher along with the transient increase in σ_{n-eff} . Moreover, with the occurrence of bulk melting, melts could be extruded from the sample assembly. These processes may explain the observed increase in the shortening rate. The following sentence has been added in the revised manuscript to clarify this point.

“Wear could be enhanced along with the transient increase in σ_{n-eff} , and the melt extrusion from the slip zone may come with bulking melting. The enhanced rates of losing wear and melting products from the slip zone may also explain the observed changes in the shortening rate...” (Line 206–208)

(10) Conclusion; I think it is better mentioning about the low effective normal stress observed in natural seismogenic fault zones (e.g., Audet et al., 2009; Peacock et al., 2011) to increase the applicability and impact of this result.

[Our response] It's a good point. The following sentences have been added in lines 235 to 238.

“Moreover, no matter how high or low the σ_{n-eff} might be in natural fault zones (the σ_{n-eff} in subduction fault zones might be as low as those in our experiments; Audet et al., 2009; Peacock et al., 2011), the much higher degree of fault roughness in nature may result in more intense wear due to stress concentration and higher amount of water trapped within the faults, both of which may facilitate the transient TP.”

(11) Extended Data Figure 2; I could not understand why the experiment with N₂ gas shows TSW as same as the one with water, despite there must be a large difference in thermal properties (e.g., thermal expansivity) and viscosity. Some explanation or model calculation must be needed.

[Our response] We plotted viscosity, thermal expansivity, and compressibility of H₂O (dotted lines) and N₂ (thin solid lines) against temperature (from 20 to 1200 °C) in the cases of pressure ranging from 10 to 50 MPa in the following figure. It is clear that the differences in these three parameters between H₂O and N₂ become minor at temperatures higher than ~500 °C. At the onset of the TSW, the measured temperature at the location of the thermocouple (~ 15 mm away from the center) is as high as 591–851 °C. Given such high-temperature rise, those three parameters of N₂ become almost identical to those of H₂O. It is thus not surprising to observe quite similar TSW.

To clarify this point, we have added the following sentence beneath the revised Supplementary Figure S2.

“The minor differences in thermal expansivity, compressibility and viscosity between H₂O and N₂ at temperatures higher than ~500 °C may explain the quite similar TSW in the cases of these two pore fluids.”

Reference:

Audet, P., Bostock, M. G., Christensen, N. I., and Peacock, S. M.: Seismic evidence for overpressured subducted oceanic crust and megathrust fault sealing, *Nature*, 457, 76–78, <https://doi.org/10.1038/nature07650>, 2009.

Simon M. Peacock, Nikolas I. Christensen, Michael G. Bostock, Pascal Audet; High pore pressures and porosity at 35 km depth in the Cascadia subduction zone. *Geology* 2011; 39 (5): 471–474. doi: <https://doi.org/10.1130/G31649.1>

[Our response] Thanks! We have quoted these two excellent papers.

Reviewer #3 (Remarks to the Author):

Review

(1) The manuscript by Yao et al. reports the results of high-speed friction experiments on the dolerite samples sheared under the condition of fluid pressure applied. The experiment configurations are novel to simulate the earthquake slip at depths more realistically. The authors also introduced a new “pitted sample” configuration with some holes to trap some fluid at the slip interface. They observed very drastic and sharp weakening and concluded with the integration of microstructure observation and numerical modeling that it is due to local thermal pressurization due to the transient local shear zone developed due to wear and melt products during the shear. Their techniques are novel and essential for advancing our understanding of earthquake slips and processes. However, due to the complexity of the experiment configuration and complicated kinetics and deformation processes, it is very challenging to understand what is happening during the test. I have concerns about some descriptions and interpretations, primarily due to the lack of critical information. Below, I made comments separately for friction experiments, microstructure observation, and numerical simulations.

[Our response] We sincerely thank the reviewer for the constructive comments and suggestions. The reviewer paid close attention to the details of friction experiments and numerical modeling, many of which were described in our initial draft, but later were shortened or deleted when we tried to meet the length requirement of *Nature Communications*. We apologize for the unclear description of the methodology in the submitted manuscript. In this revised version, we tried to address all the comments of the reviewer.

<Friction Experiments>

(2) Looking at the friction coefficient and temperature evolution shown in Extended data figure 2, I am not sure what the effective stress means in this configuration. As it is said in Lines 55-56, the samples “initially have open boundaries for fluid flow.” Clearly, confining pressure is equal to pore fluid pressure in this sample configuration.

[Our response] Yes, confining pressure is equal to pore fluid pressure in our sample configuration. By saying effective normal stress (σ_{n-eff}), the point is that *the net axial loading on the slip surface is not equal to the axial force applied by the air actuator*

(part 12 in Supplementary Fig. S1a). This is because pore pressure acts on the area underneath the bottom end of the lower loading piston, adding a counteracting downward force that could reduce the axial loading.

The axial force due to pore pressure (F_{pp}) can be roughly estimated by the pore pressure multiplied by the corresponding area or, more precisely, can be calibrated by loading/unloading cycling tests, in which we can determine the F_{pp} at the initiation of the lower piston moving upward and downward (see Fig. (a) below; Ma et al., 2014). By conducting a series of calibration tests under different pore pressure, we can well describe the interaction between axial force and pore pressure (see Fig. (b) below; Ma et al., 2014). The difference in F_{pp} between the upward and downward cases is twice the extra force due to the O-ring friction. As shortening means the lower piston moving upward, we use the relation between F_{pp} and P_p determined for the upward case to calculate σ_{n-eff} in the data processing.

The effective normal stress (σ_{n-eff}) on the slip surface (the area is A) is then:

$$\sigma_{n-eff} = (F_a - F_{pp}) / A = [F_a - (177.9 * P_p \text{ [MPa]} + 90.5)] / A,$$

where F_a is the axial force applied by the air actuator. The recorded data of F_a and P_p were used to calculate σ_{n-eff} in the data processing.

Based on this comment and the comment (5) below, we have added a few sentences in the Method section to clarify the above-mentioned information. (Lines 272–277)

*“For the sample configuration we used, the pore pressure could impose a downward force (F_{pp}) that reduces the net axial loading on the slip surface. In the data processing, we used the recorded data of bulk P_p and axial force from the air actuator (F_a), and the calibrated relation between F_{pp} and P_p to determine σ_{n-eff} ($= (F_a - F_{pp}) / A = [F_a - (177.9 * P_p \text{ [MPa]} + 90.5)] / A$, where A is the area of the slip surface; Ma et al., 2014). The changes in bulk P_p only brought about small deviations (< 0.3 MPa) from the initial σ_{n-eff} during the experiments.”*

(3) I believe N2 pressure is kept constant at the upper part of the pressure vessel during the test to keep the water or N2 Pp constant, correct? It is said in Lines 231-232 that any changes in local pressure (I’m not 100% sure what “local pressure” means) could be significantly buffered due to the high compressibility of nitrogen and the relatively large volume of water in the vessel. However, the authors observed an increase in Pp on the transducer located at the lower part of the pressure vessel. Does that mean the

pore pressure pressurization system can't catch up with rapid sample responses such as P_p change (and sample dilation/compaction) within the shear zone during the high-speed shear? Or do you have a closed system ("undrained") that does not intend to control the pore pressure constant during the test?

[Our response] In the previous Extended Data Fig. 1b, we didn't specify the pressure regulator we used. It was a high-precision pressure-reducing regulator (not a back-pressure regulator). The reducing regulator can maintain the downstream pressure (P_{p_down}) constantly at a desired value that is lower than the upstream pressure (P_{p_up}). However, for a given set point (i.e., a given set position of the pressure setting screw), in principle, the poppet valve inside the regulator will be closed if the P_{p_down} increases, making the downstream side become a closed chamber (see how the regulator may work in the schematic diagram shown below).

This is confirmed by a calibration test below, where two syringe pumps were connected to the inlet and outlet ports of the reducing valve, with the P_{p_up} and P_{p_down} being monitored using two pore pressure transducers. We kept the P_{p_up} at 30 MPa and adjusted the pressure-reducing regulator to get the P_{p_down} of 25 MPa, during which the downstream pump was stopped (i.e., the downstream side acted as a closed chamber). With the fixed set point of the regulator, we then ran the downstream pump to increase the pressure step by step towards 35 MPa while the upstream pump was stopped. The P_{p_up} remains constant whilst the P_{p_down} increases, suggesting no interaction between the upstream and downstream in such a case. Thus, in our experiments, the pressure vessel is indeed a closed chamber when the bulk pore pressure increases.

In the revised manuscript (the Method section), we have briefly described that the pressure vessel is a closed system during the experiments. (Lines 268–270)

(4) Also, pore pressure measurement on the pressure transducer was not shown for the entire duration of the experiments (Extended Data Figure 2; c or d). Why?

[Our response] We just thought the pore pressure data in the later stages of experiments (> 8 m slip) do not provide very useful information about our main concern—the TSW.

Since the reviewer is curious about this, we have added the data in the revised Supplementary Fig. S2d.

(5) Related to the above question, how is the experiment's friction coefficient calculated? I believe you divide shear stress, which is calculated from the measured torque, by effective normal stress, but do you assume the effective normal stress is constant during the shear? The effective normal stress changes during the shear as the pore pressure changes. I understand it is probably impossible to quantify the effective normal stress within the shear zone because of the unknown pore pressure there; however, the measured pore pressure (called bulk pore pressure in the manuscript) also changes. Do you account for the bulk pore pressure change to calculate effective normal stress? The effect is probably small, but it would be good to know how the reported values are calculated.

[Our response] As described in response to the reviewer's comment (2), we use the recorded data of axial force (applied by the actuator) and bulk P_p throughout the experiment to calculate σ_{n-eff} in the data processing.

Just as the reviewer speculated, the effect of bulk P_p changes on σ_{n-eff} is small. The change in bulk P_p only brought about a small derivation from the initial σ_{n-eff} (< 0.3 MPa, which corresponds to the σ_{n-eff} change due to the observed maximum P_p change of 1.6 MPa).

(6) For the tests with pore fluid of water, I can see relatively quick recovery followed by a drastic drop in frictional coefficient at a displacement of ~ 9 m and ~ 12 m (Extended data Figure 2). The caption says, "the large fluctuation in friction during the re-strengthening stage in the cases of pore water pressure (LHV968 and LHV969) is probably caused by the complicated fracturing, wear-induced sealing, and bulk melting processes in the presence of water." The response in both tests is very similar; thus, I wonder if some systematic mechanism causes the fluctuation.

[Our response] Because the large fluctuation can be seen in the presence of water rather than nitrogen, we believe the main cause is probably the complicated fracturing process promoted by quenching in water. We have added a few words "*(probably promoted by quenching in the surrounding cool water)*" after "the complicated fracturing..." in the caption of the Supplementary Fig. S2.

(6, Continued) Also, the temperature evolution in LHV969 and LHV965 becomes similar after ~ 15 m slip. Do you think that's the time bulk melting occurs?

[Our response] Yes, we agree that the convergence of the temperature curves after ~ 15 m slip may suggest the occurrence of bulk melting. One explanatory sentence has been added in the caption of the Supplementary Fig. S2.

(6, Continued) The temperature of LHV 968 after ~ 15 m is not shown in Figure. Is that because the thermocouple was damaged? Some explanation would be great to have.

[Our response] Yes, we believe the thermocouple was badly worn after 15 m slip in the run LHV968. To avoid the vague information, we plot the entire temperature data of the run LHV968 instead in the revised Supplementary Fig. S2, and explain that the thermocouple was probably badly worn after 15 m slip in the caption.

(6, Continued) By the way, it is hard to distinguish the temperature data shown on panel d of Extended Data Figure 2. Could you use the solid line with different colors shown on panel c?

[Our response] Apologize for the vague information in the figure. We have made changes to the figure as suggested.

(7) Extended data figure 2d: In the figure caption, it is mentioned that “the overall significantly lower temperature in the case of 10 MPa water should be attributed to the energy sink associated with the water phase change at the boiling point, which is 311 °C at $P_p = 10$ MPa (much lower than 591 °C)”. I don’t fully understand the argument if you are talking about the difference between LHV969 ($P_p = 10$ MPa with water) and LHV 968 ($P_p = 25$ MPa with water). The difference in temperature between LHV969 ($P_p = 10$ MPa with water) and LHV968 ($P_p = 25$ MPa with water) is likely attributed to the peak frictional coefficient observed at the beginning of the shear.

[Our response] We agree that the higher peak friction coefficient in the run LHV968 may substantially contribute to the observed higher temperature rise. However, by comparing the results from the three tests (LHV965, 968 and 969; Supplementary Fig. S2), we can observe that the temperature at the displacement of 1~1.6 m is lowest in the test LHV969 ($P_p = 10$ MPa with water), although the overall friction in this test is even slightly higher than that in the test LHV965 ($P_p = 10$ MPa with air). This is why we argue that the energy sink associated with the water phase change in the test LHV969 may play a role in affecting the temperature rise.

As this issue could be controversial, we simply delete the relevant sentences in the revised manuscript.

(8) Transient sharp weakening (TSW) accompanies by an abrupt increase in the rate of bulk pore pressure rise, and I noticed that P_p raise is relatively large; for example, in LHV968, the bulk pore pressure rise reaches ~1.7 MPa, which is about 7 % of the pore pressure (25 MPa). Have you calculated how much volume change is required to cause such a rise in a short period? As I mentioned earlier, I’m unsure if the pore pressure is controlled at constant pressure at the upper side of the pressure vessel.

[Our response] As clarified in response to the reviewer’s comment (3), the pressure vessel is a closed system during our experiments. The total volume of nitrogen, water, rock sample and metal parts (loading columns and sample holders) thus maintains

constant in the tests. The initial volume of water added into the vessel is about 200 ml. There is only small shortening (< 0.3 mm for the tests shown in Figs. 2b–d) during the TSW, so the increase in the volume of the lower loading column inside the vessel is less than ~0.5 ml. The volume change of the rock sample associated with wear and melting may also be very small. The major contribution to the bulk pore pressure rise should be the thermal expansion of pore water, especially in the vicinity of the slip surfaces. However, we think it is an arduous task to estimate the volume change of water because of the rapid temperature changes and the transient pressure gradient in the vessel. Boyle's law would be inapplicable, and the involved physico-chemical processes are too complicated to reproduce by numerical models.

(9) The authors also described a possible correlation between TSW and axial shortening in Lines 106 and 110. However, no interpretation was provided on it. What causes shortening? A question is whether wear/melt products of dolerite are kept in place during/after the shear or if some are floating or deposited somewhere in the pressure vessel. Because the sample configuration cannot confine the sample, losing some materials leads to shortening.

[Our response] We thought lines 106 and 110 are in the result section, so we didn't provide an interpretation there. Lines 171 to 174 in the previous manuscript briefly explain why TSW and axial shortening may correlate. As the reviewer speculated, wear/melt products of dolerite could lose from the slip zone, causing axial shortening. In the revised manuscript, we have added a few sentences to explain more about the possible correlation between the TSW and axial shortening.

“The next slip strengthening is due to the failure of sealing and melt patches (the escape of pressurized water may increase σ_{n-eff} significantly) as well as the initiation of bulking melting (stage 4 in Fig. 5). Wear could be enhanced along with the transient increase in σ_{n-eff} , and the melt extrusion from the slip zone may come with bulking melting. The enhanced rates of losing wear and melting products from the slip zone may explain the observed changes in the shortening rate.”

(Lines 204 to 209)

(10) Temperature data of the experiments at effective normal stress = 6 or 10 MPa were not presented. Are they not measured? Just not presented? These are test conditions where you compared pitted samples vs. flat samples. I am not sure how the pitted samples contributed to the pore pressure evolution or TSW because TSW is observed on the flat samples at effective normal stress = 3 MPa.

[Our response] We only monitored temperature evolution in several selected experiments. These include four tests conducted at σ_{n-eff} =3 MPa (LHV964, 965, 968 and 969), one test conducted at σ_{n-eff} =6 MPa (LHV1022), and another test conducted at σ_{n-eff} =10 MPa (LHV1021). Unfortunately, we didn't measure temperature in the experiments conducted on the “pitted samples” at σ_{n-eff} =6 MPa and 10 MPa. However, as friction acts as heat source, we may expect a transient drop or a slower increase in temperature when

the TSW occurs in the “pitted sample” experiments at $\sigma_{n-eff}=6$ MPa and 10 MPa. Here we show the temperature data from the “flat sample” experiments below (blue solid and dashed lines); the thermocouple might be badly worn in the run LHV1022).

What hinder us from measuring temperature in all the experiments include the high cost of the experiment due to thermocouple damage after every test, and the time-consuming procedures in setting up the thermocouple (e.g., drilling a $\Phi 2$ hole, and fixing the sheathed thermocouple with high-temperature waterproof cement, etc).

The reviewer also asked about how the pitted samples contributed to the pore pressure evolution. We think this is already illustrated in Fig. 2c and d (cp. the solid and dashed green lines). With the pitted slip surfaces, we could observe a spike in pore fluid pressure that is concurrent with the TSW.

(11) For “pitted sample” tests, do both stationary and rotary blocks have pits or only one block? Where are the pits located? In Extended data figure 1d, I can see one pit at the center and the other 8 holes, but are they located at a distance from the center? If so, how far are they? Please clarify.

[Our response] In the “pitted sample” tests, only the stationary block has pits. One pit is located in the center and the other 8 pits are located 14 mm from the center. We have added one sentence to clarify this in the caption of the Supplementary Fig. S1.

(12) The location (distance from the center) of the thermocouple in the experiment set was not mentioned clearly. I believe it is located 15 mm from the center based on the information given for the numerical simulation. Please clarify.

[Our response] Yes, the tip of the sheathed thermocouple is located 15 mm away from the center of the slip surface. We have added some words to clarify this. (Line 77).

<Microstructure observations>

(13) The microstructures were presented for the test LHV1103, with a total slip of 4.5 m in Figure 3 a-c and for LHV968, with a slip of 20 m in Extended data figure 4. No

information on the location of each image. There is some inset for Figures 3b and 3c, but each image's location and direction are unclear (e.g., distance from the center; slip parallel or slip perpendicular). Also, the mechanical data of this experiment (LHV1103) was not presented. I understand the test conditions of normal stress, P_p , and slip rate are the same as LHV968, but I'm not sure the mechanical response is the same. Please add some description in the text or present the mechanical data of LHV 1103.

[Our response] *We have made substantial revisions to Fig. 3 and Supplementary Fig. S5 in accordance with the reviewers' comments (another reviewer also commented on this point). The information about the locations and directions of all the close-up images can be found in the revised figures.*

With regard to the mechanical data of the run LHV1103, the first reviewer also commented on this point. Basically, the pre-peak friction evolution is complicated and the peak friction appears at a much larger displacement in the run LHV1103, but *the test seems to be terminated at the onset or initial stage of the transient sharp weakening.* In the revised manuscript, the data of the run LHV1103 is presented in Supplementary Fig. S4.

(14) What is the expected melting temperature of your dolerite sample?

[Our response] We performed simultaneous thermal analysis (Thermogravimetric analysis–TG + Differential scanning calorimetry–DSC) on the crushed dolerite powder (grain size < 75 μm). It turns out that the melting temperature is about 1060 °C (see DSC results below). However, this is only the melting temperature under the dry condition, and it could be a few hundred lower in the presence of pore water with the help of comminution and wear. Based on the measured temperature data (Supplementary Fig. S2d) and the postmortem microstructures (continuous molten layer developed in the runs LHV965, 968 and 969; see an example in Supplementary Fig. S5b), we think the melting temperature of dolerite during the experiments might be 850–950 °C.

<FEM Modeling>

(15) It is unclear why you have three ring seal zones vs. one seal zone for the modeling setup. I don't see any discussion on the difference between single and three seal zones. I don't also know what motivated authors to set up three ring seal zones. Are they based on some microstructure observations?

[Our response] Yes, the ring-shaped sealing zones in the numerical models are constructed based on the postmortem microstructures. As shown in Fig. 3a and Supplementary Fig. S5a, continuous ring-shaped slickensides were developed in overall rough and fractured slip surfaces. Those slickensided regions with local melting and wear products might act as sealing zones during the experiments.

However, we cannot well constrain the number, the exact location and geometry of the sealing zones, and their evolution with time during the experiments. As first-order assessments of the TP processes in our experiments, we simply consider two cases, one or three ring-shaped sealing zone(s), in the modeling. We have added a few sentences in both Results (lines 155–160) and Method (lines 331–333) sections to discuss the setting of the geometrical model in the modeling.

(16) What is the time = 0 of the modeling? At the beginning of TSW?

[Our response] The time of peak friction is 0 s in the modeling. We have described this in the revised manuscript. (Line 176)

Figure 3:

(17) In panel d, are the data for Dry, and $k_{bsz}=k_{seal}=10^{-10}$ and 10^{-11} shown here? I don't see them, but they may be behind the result of $k_{bsz}=k_{seal}=10^{-12}$.

[Our response] Yes, the curves for $k_{bsz} = k_{seal} > 10^{-12} \text{ m}^2$ overlap in the Fig. 3d. I have added a few words to explain this. (Lines 170 to 171)

(18) In pane e, I don't see the data for $k_{bsz} = 10^{-8}$ to 10^{-12} ...

[Our response] Again, the curves overlap. We actually mentioned this in our previous manuscript “...the μ_{eq} curves overlap for the cases of $k_{bsz} = 10^{-8}$ – 10^{-13} m^2 ...”.

(19) How about the boundary conditions at the end of the samples? The sample is immersed in the fluid (water) in the pressure vessel. Do you have a boundary condition of no flow or constant pressure or anything else at the periphery of the shear zone?

[Our response] The measured bulk pore pressure rise ranged from ~0.15–1.6 MPa in the experiments. For simplicity, the pressure at the periphery of the shear zone is assumed to be constant at 25 MPa in the modeling. We have described this in the Method section (Lines 333 to 335).

(20) The description of the heat source setting (Lines 258-262) was unclear. The authors set up a localized line heat source at the center of the shear zone constrained by the

time-dependent temperature. The temperature is a function of the sample radius (distance from the sample center), $T(r,t)$ is proportional to r , and $T(r = 15 \text{ mm}, t)$ is constrained by the thermocouple measurements during the friction experiments. If you assume $T(r,t)$ is proportional to r , does that mean that $T(r = 0, t)$ is fixed (at room temperature?) and stays constant during the shear? If that's the case, I don't think this is an appropriate setting. At the center of the sample, heat should not be generated because the shear velocity is zero, but that is not the same as constant temperature. So, I'm concerned that the heat source is constrained by the temperature. Please clarify this.

[Our response] This is another very constructive comment. We agree that it is inappropriate to assume a fixed temperature at $r = 0$. Here we first explain why we think it is reasonable to assume a linear relation between $T(r, t)$ and r . Using the geometrical model built in our TP modeling, if we prescribe a boundary heat source in the center of the shear zone and only calculate temperature rise by solving the heat diffusion equation, the estimated temperature along the radial direction roughly shows a linear increase with r at the given time ($T(r, t)$ is plotted against r in the figure below; the results are for $t = 0$ to 0.9 s, with a t increment of 0.05 s).

Then we can make a simple but reasonable change to the assumption of $T(r, t)$ versus r — $T(r, t)$ is proportional to r at $r \geq 1 \text{ mm}$, and $T(r, t)$ at $r < 1 \text{ mm}$ is equal to $T(r = 1, t)$. This means that $T(r, t)$ at $r < 1 \text{ mm}$ could evolve with t , although it is spatially constant.

We have modified the setting of $T(r, t)$ versus r in the models as described above and re-performed all the numerical modeling. The new results are quite similar to previous ones. We have presented these in Fig. 4 of the revised manuscript. Some descriptions have also been modified in the Method sections (lines 306–308).

<Minor comments>

(21) Line 95: “Extended Data Figs. 1a-c” should be “Extended Data Figs. 2a-c”?

[Our response] Yes, we have corrected the mistake.

(22) Line 111: “weld” should be “welded”

[Our response] Thanks! “weld” has been replaced with “welded”.

REVIEWERS' COMMENTS

Reviewer #3 (Remarks to the Author):

I appreciate the authors' responses to address concerns in the previous review. The revised manuscript was significantly improved and very close to publication. I have a couple of comments, which can be addressed by the authors quickly.

"Equivalent friction coefficient, μ_{eq} " (defined in Lines 298-302) was used, but the meaning of "equivalent" here is different from other "equivalent" parameters such as equivalent slip velocity and equivalent displacement. Equivalent slip velocity and equivalent displacement were used as a single value to represent various values in a rotary shear friction configuration where slip velocity and displacement within the sample are not constant but increase with the distance from the cylindrical axis of the sample. This study assumes that any friction coefficient values (equivalent friction coefficient or intrinsic friction coefficient) are constant within the sample. "Equivalent friction coefficient" is typically called as "effective" friction coefficient or "apparent" friction coefficient, which includes the effect of pore pressure. To avoid confusion, I suggest authors consider calling out the parameter in a different way.

Lines 189 and 191: "bulking melting" should be "bulk melting".

Reviewers' comments and our responses

In this response letter, we listed the reviewer's comments on our revised manuscript (NCOMMS-22-14093A) in black and wrote our responses in blue following each comment. The manuscript has been modified accordingly.

Reviewer #3 (Remarks to the Author):

I appreciate the authors' responses to address concerns in the previous review. The revised manuscript was significantly improved and very close to publication. I have a couple of comments, which can be addressed by the authors quickly.

(1) "Equivalent friction coefficient, μ_{eq} " (defined in Lines 298-302) was used, but the meaning of "equivalent" here is different from other "equivalent" parameters such as equivalent slip velocity and equivalent displacement. Equivalent slip velocity and equivalent displacement were used as a single value to represent various values in a rotary shear friction configuration where slip velocity and displacement within the sample are not constant but increase with the distance from the cylindrical axis of the sample. This study assumes that any friction coefficient values (equivalent friction coefficient or intrinsic friction coefficient) are constant within the sample. "Equivalent friction coefficient" is typically called as "effective" friction coefficient or "apparent" friction coefficient, which includes the effect of pore pressure. To avoid confusion, I suggest authors consider calling out the parameter in a different way.

[Our response] We sincerely thank the reviewer for the suggestion on the term used to describe the "friction coefficient" that incorporates the weakening effects of thermal pressurization. We totally agree with the reviewer's comments above.

In accordance with the reviewer's suggestion, we replace the term "equivalent friction coefficient" with "apparent friction coefficient" throughout the manuscript (Lines 162 and 301, and the Y-axis titles of Figure 4 and Supplementary Figure S7). Also, " μ_{eq} " appearing in several places have been replaced with " μ_{ap} " (Lines 162, 166, 173, 301, 303 and 305).

(2) Lines 189 and 191: "bulking melting" should be "bulk melting".

[Our response] Apologize for the spelling mistakes. We have made corrections as suggested (Lines 189 and 191).